# THINK-WHILE-GENERATING: ON-THE-FLY REASONING FOR PERSONALIZED LONG-FORM GENERATION

**Chengbing Wang**[1]  **Yang Zhang**[2]*  **Wenjie Wang**[1]  **Xiaoyan Zhao**[3]  **Fuli Feng**[1]
**Xiangnan He**[4]*  **Tat-Seng Chua**[2]
[1]University of Science and Technology of China, [2]National University of Singapore
[3]The Chinese University of Hong Kong
[4]National Engineering Laboratory for BITA, University of Science and Technology of China
wwq197297@mail.ustc.edu.cn

## ABSTRACT

Preference alignment has enabled large language models (LLMs) to better reflect human expectations, but current methods mostly optimize for population-level preferences, overlooking individual users. Personalization is essential, yet early approaches—such as prompt customization or fine-tuning—struggle to reason over implicit preferences, limiting real-world effectiveness. Recent "think-then-generate" methods address this by reasoning before response generation. However, they face challenges in long-form generation: their static one-shot reasoning must capture all relevant information for the full response generation, making learning difficult and limiting adaptability to evolving content. To address this issue, we propose **FlyThinker**, an efficient "think-while-generating" framework for personalized long-form generation. FlyThinker employs a separate reasoning model that generates latent token-level reasoning in parallel, which is fused into the generation model to dynamically guide response generation. This design enables reasoning and generation to run concurrently, ensuring inference efficiency. In addition, the reasoning model is designed to depend only on previous responses rather than its own prior outputs, which preserves training parallelism across different positions—allowing all reasoning tokens for training data to be produced in a single forward pass like standard LLM training, ensuring training efficiency. Extensive experiments on real-world benchmarks demonstrate that FlyThinker achieves better personalized generation while keeping training and inference efficiency.

## 1 INTRODUCTION

Large language models (LLMs) have achieved significant advances through preference alignment techniques, enabling them to produce outputs that more closely reflect human expectations (Ouyang et al., 2022; Rafailov et al., 2023). Yet, common preference alignment (Rafailov et al., 2023; Schulman et al., 2017) primarily focuses on population-level preferences, often overlooking the diverse characteristics and nuanced needs of individual users (Guan et al., 2025; Xie et al., 2025; Li et al., 2024; Zhao et al., 2025b; Qiu et al., 2025a;b; Zhao et al., 2026; Zhang et al., 2025c). These limitations inevitably curb user satisfaction and hinder the more widespread adoption of LLMs in everyday life (Guan et al., 2025; Wu et al., 2025; Zollo et al., 2025; Zhao et al., 2025a; Wang et al., 2025a; Shi et al., 2024; Wang et al., 2025b). To bridge this gap, personalization has emerged as an indispensable component of LLM alignment—an essential step toward integrating broad, generalized utility with the fine-grained, personalized adaptability required for truly user-centric AI (Sorensen et al., 2024).

Existing LLM personalization methods typically rely on either customizing user-specific prompts via RAG (Qian et al., 2024; Mysore et al., 2023b; Zhang et al., 2025a), or fine-tuning the model (Zhang et al., 2025d; Chen et al., 2025; Kim et al., 2025c) to produce personalized outputs. While straightforward, these approaches often struggle in real-world scenarios where user

---

*Corresponding author: {zyang1580, xiangnanhe}@gmail.com

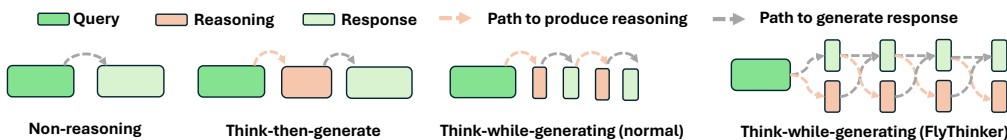

Figure 1: Different reasoning–generation paradigms. Non-reasoning outputs responses directly. Think-then-generate performs reasoning first, then responds. Standard think-while-generating interleaves reasoning and output sequentially, causing inefficiency. FlyThinker enables reasoning and generation to proceed in parallel.

interests are not explicitly stated but are only implicitly reflected in historical behavior. Such methods lack a mechanism to reason over implicit user preferences, potentially resulting in suboptimal personalization. Aiming at addressing this limitation, a new "think-then-generate" paradigm for LLM personalization has emerged (Luo et al., 2025b; Kim et al., 2025a; Li et al., 2025b; Salemi et al., 2025c): rather than directly generating the final response, the LLM first produces an in-depth analysis of the user's preferences based on their input, and then generates responses aligned with this analysis. By leveraging the model's reasoning capabilities, this approach can potentially capture implicit user interests more effectively, enabling more accurate personalization.

Despite its promise, the "think-then-generate" paradigm faces critical challenges in long-form personalized generation. In this paradigm, a static one-shot reasoning must capture all relevant information for the entire response, creating long-range dependencies that are difficult to model and learn. Moreover, it rigidly applies this one-step reasoning to guide all subsequent outputs, failing to adapt to dynamic changes in content — long-form creation by users often involves ideas that evolve based on what they have already created. To address these, we propose adopting a **"think-while-generating"** paradigm, as illustrated in Figure 1, where LLM reasoning and answer generation are interleaved, with each reasoning step focusing on guiding a small segment of the response. This design better aligns with the natural progression of user thinking during creation, reduces learning complexity, and thereby potentially enables more effective personalization.

To implement the think-while-generating paradigm, a key challenge is efficiency: frequent reasoning across response segments substantially increases both training and inference time as the number of segments grows. Latent reasoning (Hao et al., 2024) offers a more efficient alternative to standard chain-of-thought (CoT) approaches (Wei et al., 2022) by compressing the reasoning path into a few hidden-state tokens, avoiding the overhead of generating lengthy CoT sequences. However, directly applying existing latent reasoning methods remains inefficient. Current approaches (Tang et al., 2025; Zhang et al., 2025b) typically use the same LLM for reasoning and generation, forcing each output token generation to wait for all preceding reasoning tokens. Non-reasoning training can be accelerated by feeding all ground-truth tokens simultaneously, enabling parallel optimization. With reasoning tokens, this parallelism is lost, as the model must wait for the reasoning outputs, slowing down training. For inference, the time cost also increases with the total number of reasoning tokens. To solve the problem, the key is to keep the parallelism.

To this end, we propose **FlyThinker**, an efficient "think-while-generating" framework for LLM personalization with latent reasoning. FlyThinker enables parallel training and inference by maintaining a separate reasoning model that continuously generates token-wise latent reasoning based on the query and the tokens produced so far. This reasoning is fused into the generation model to guide token prediction, allowing the generation process to incorporate dynamic reasoning. Importantly, reasoning tokens at different positions do not have direct sequential dependencies—they depend only on the query and the already generated tokens—allowing reasoning to run in parallel during training when all ground-truth tokens are input for use in a teach-forcing manner. Once reasoning at all positions is computed in parallel, the generation model can also operate in parallel to predict all ground-truth tokens during training. During inference, the two models operate in a staggered manner: while the generation model predicts the current token, the reasoning model prepares reasoning for the next prediction. This design allows FlyThinker to implement token-level think-while-generating efficiently, avoiding sequential bottlenecks while ensuring that each prediction is guided by up-to-date reasoning.

In summary, the main contributions are three-fold. **Firstly**, we put forward the concept of the **think-while-generating** paradigm for personalized long-text generation. **Secondly**, we propose FlyThinker, a novel framework that integrates a parallel reasoning model, achieving efficient token-

level think-while-generating. **Lastly**, we conduct extensive experiments demonstrating that Fly-Thinker delivers substantial improvements in both personalization quality and generation efficiency over strong baselines.

## 2 TASK FORMULATION

This work studies personalized long-form generation, where the LLM must infer user preferences from historical data and produce responses that align with these preferences for new queries. Formally, let $\mathbb{D}$ denote the collected dataset, and let $(h, x, y) \in \mathbb{D}$ represent a user-specific sample, where $x$ is the query, $y$ is the ground-truth response, which naturally is long (*e.g.*, a full movie review written by the user), and $h$ is the historical record consisting of the user's past (query, response) pairs. The goal is to leverage $\mathbb{D}$ to adapt the LLM such that, given $(h, x)$, it generates a response $\hat{y}$ consistent with the user's preferred response $y$.

Since user preferences are only implicitly encoded in historical data, it is better for the LLM to involve a reasoning process to infer them before generating the final response. Existing work typically adopts the "think-then-generate" paradigm to involve the reasoning, which can be expressed as:

$$\textit{Think-then-Generate:} \quad (h, x) \xrightarrow{LLM} r \xrightarrow{LLM} \hat{y}, \tag{1}$$

where $r$ denotes the reasoning trace. However, in long-form generation, this paradigm often suffers from low learning efficacy and misalignment with the inherently dynamic nature of long-form writing—users' ideas evolve as the response unfolds. To address this, we consider a think-while-generating paradigm: after producing a partial response (a token in this work), the model generates a new reasoning step to guide the next segment of text. This can be formulated as:

$$\textit{Think-while-Generating:} \quad (h, x) \xrightarrow{LLM} r \xrightarrow{LLM} \hat{y}_1 \xrightarrow{LLM} r \xrightarrow{LLM} \hat{y}_2 \ldots, \tag{2}$$

where $\hat{y}_k$ denotes the $k$-th token of the generated response. This work focuses on how to achieve this "think-while-generating" paradigm efficiently.

## 3 METHODOLOGY

In this section, we introduce FlyThinker, covering its overview, architecture, training, and inference.

### 3.1 OVERVIEW

To efficiently achieve "Think-while-Generating", we identify a key challenge in existing approaches: using the same model for both generation and reasoning. In this setting, generating the next response token or reasoning step must wait for the completion of previous reasoning, leading to unacceptable delays. To address this, we propose FlyThinker, which parallelizes response generation and reasoning by employing two separate models for these tasks, as shown in Figure 2. At each step, while the generation model produces a response token, the reasoning model can simultaneously generate a new reasoning segment, which then guides the next step of response generation. Formally,

$$
\begin{aligned}
\textit{Generation:} \quad & (h, x) \xrightarrow{(h, x; \hat{y}_{<1} + r_{<1})} \hat{y}_1 \xrightarrow{(h, x; \hat{y}_{<2} + r_{<2})} \hat{y}_2 \ldots, \\
\textit{Reasoning:} \quad & (h, x) \xrightarrow{(h, x, \hat{y}_{<1})} r_1 | \xrightarrow{(h, x, \hat{y}_{<2})} r_2 | \ldots,
\end{aligned}
\tag{3}
$$

where $r_t$ denotes the new reasoning results at step $t$, and $\hat{y}_{<t}$ and $r_{<t}$ represent the response and reasoning generated before step $t$. This obviously ensures that inference can be performed efficiently.

Based on this parallel model design, we further break the direct sequential dependence between reasoning tokens while ensuring that each reasoning step depends only on previously generated response tokens, as indicated by $r_{t-1} | \xrightarrow{(h, x, \hat{y}_{<t})} r_t$, meaning $r_t$ depends only on $(h, x, \hat{y}_{<t})$. This blocking design enables parallel training: during training, all reasoning steps can be generated in parallel by providing the reasoning model with the full ground-truth response sequence, improving training efficiency, similarly for the generation side. For generation, each step depends on both the previously generated reasoning and response, as indicated by $\hat{y}_{t-1} | \xrightarrow{(h, x, \hat{y}_{<t} + r_{<t})} r_t$, meaning $\hat{y}_t$ depends on $(h, x, \hat{y}_{<t} + r_{<t})$, ensuring that dynamic reasoning is truly considered for the generation.

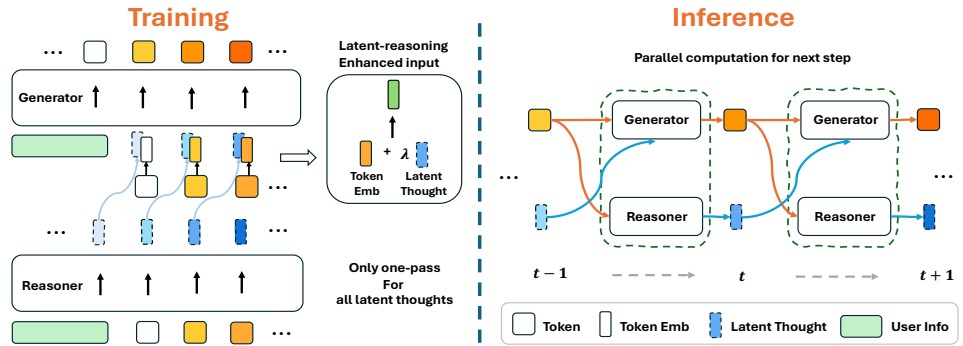

Figure 2: An illustration of **FlyThinker**, an efficient "think-while-generating" framework for LLM personalization with latent reasoning.

## 3.2 MODEL ARCHITECTURE

We enable parallel response generation and reasoning by introducing two separate models: a Reasoner, which produces **latent reasoning**, and a Generator, which leverages it for response generation. These models are organized in a parallel architecture—while the Generator produces a response token, the Reasoner simultaneously generates a new reasoning segment to guide the next step of response generation, as shown in Figure 2. We next elaborate on the two models to explain the parallel architecture.

**Reasoner.** The Reasoner ($R$) is an LLM designed to produce a **latent** reasoning token at each step of response generation. At step $t$, it takes the query and the previously generated response as input to produce a new latent reasoning token—similar to how humans update their thinking during the writing process based on newly written content. Following common practice in existing works, the latent reasoning token at each step $t$ is extracted from the hidden state of the top layer of the Reasoner. Formally,

$$r_t = R_\theta^{(-1)}(h, x; \hat{y}_{<t-1})[-1], \tag{4}$$

where $R_\theta^{(-1)}(\cdot)[-1]$ denotes the extraction of the hidden state of the last input token at the final layer of the Reasoner, $r_t \in \mathbb{R}^d$ denotes the new generated latent thought with dimension $d$, and $\hat{y}_{<t-1}$ denotes the token generate before step $t$. Notably, the Reasoner considers only the previously generated response tokens and does not explicitly include prior reasoning, breaking the direct sequential reliance between different parts of reasoning. This design is critical for enabling parallel training, which will be explained in detail later.

**Generator.** The Generator ($G$) is implemented as another LLM but enhanced by incorporating latent thoughts into the token prediction process. Instead of relying solely on discrete token embeddings, the Generator fuses latent thoughts with response embeddings at each token position, allowing reasoning signals to directly influence generation. Formally, for the $t$-th token prediction, the predicted probability is formulated as follows:

$$P(\hat{Y}_t = y_t \mid h, x, y_{<t}) = G_\phi(y_t | h, x, f(\hat{y}_{<t}, r_{<t})), \tag{5}$$

where the fuse operation $f(\cdot)$ is performed in the LLM token embedding space as

$$f(\hat{y}_{<t}, r_{<t}) = [e(y_1) + \lambda r_1, \ldots, e(y_{t-1}) + \lambda r_{t-1}], \tag{6}$$

with $e(y_1) \in \mathbb{R}^d$ denoting the embedding of token $y_1$, similar for others. The coefficient $\lambda$ controls the strength of the latent reasoning signal injected into the Generator.

## 3.3 PARALLEL TRAINING

We jointly optimize the Reasoner and Generator in an end-to-end manner, allowing the model to learn reasoning and response generation simultaneously without requiring external supervision for reasoning.

Benefiting from our "breaking sequential dependence" design (where each reasoning step does not directly rely on previous reasoning) in the Reasoner, we can achieve parallel training of reasoning

and response generation among different token positions using the teacher-forcing technique, by feeding all ground-truth tokens into the model, as in standard LLM training.

Specifically, at each training iteration, for each training example $(h, x, y)$, we input the entire $y$ into the Reasoner and perform a single forward pass to obtain the latent reasoning for all token positions. Formally,

$$r^\star = [r_1, \ldots, r_T] = R_\theta^{(-1)}(h, x; y_{<T})[-T : -1],\tag{7}$$

where $T$ is the length of $y$, $y_{<T}$ denotes all tokens of $y$ except the last one, and $R_\theta^{(-1)}(\cdot)[-T : -1]$ extracts the hidden states corresponding to the last $T$ tokens of the input sequence.

After obtaining reasoning results for all steps, the Generator predictions can be computed in a similar one-pass forward computation. Formally,

$$[P(\hat{Y}_1|h, x), \ldots, P(\hat{Y}_t|h, x, y_{<t})] = G_\phi(h, x, f(y_{<T}, r^\star_{<T}))[-T : -1],\tag{8}$$

where $G_\phi(\cdot)[-T : -1]$ extracts the predicted probability distributions of $G$ at the last $T$ token positions.

The Generator and Reasoner are then jointly optimized using the standard next-token prediction loss (denoted by $\mathcal{L}$):

$$\mathcal{L} = - \sum_{(h,x,y)\in\mathbb{D}} \sum_{t=1}^{|y|} \log P(\hat{Y}_t = y_t \mid h, x, y_{<t}).\tag{9}$$

This optimization ensures that the Reasoner is updated naturally alongside the Generator, allowing FlyThinker to capture user-specific preferences without explicit annotations or auxiliary objectives. Importantly, since all latent reasoning tokens are computed in parallel, training remains efficient, with computational costs close to standard LLM fine-tuning.

### 3.4 PARALLEL INFERENCE

For inference, we run the Generator (Equation 5) and the Reasoner (Equation 4) in parallel, enabling efficient decoding without any additional delay compared to a standard non-reasoning LLM, as illustrated in Algorithm 2. Formally, at each step:

$$\begin{aligned}
\text{(Parallel reasoning)} \quad & r_t = R_\theta^{(-1)}(h, x; \hat{y}_{<t-1})[-1], \\
\text{(Parallel generation)} \quad & \hat{y}_t \sim P(\hat{Y}_t = y_t \mid h, x, y_{<t}) = G_\phi(y_t|h, x, f(\hat{y}_{<t}, r_{<t})),
\end{aligned}\tag{10}$$

As the equations show, while the Generator produces a token, the Reasoner simultaneously generates a new latent reasoning token for the next step. This eliminates waiting, enabling response generation without time delay. Notably, the computational cost is not similarly reduced, and memory usage increases.

## 4 EXPERIMENTS

In this section, we conduct experiments to answer the following research questions: 1) **RQ1:** How does FlyThinker compare with strong baselines on personalized long-form text generation? 2) **RQ2:** What are the training and inference efficiencies of FlyThinker? 3) **RQ3:** Does FlyThinker improve generation quality across different token positions? 4) **RQ4:** How do the key architectural components of FlyThinker contribute to its overall performance?

### 4.1 EXPERIMENT SETTINGS

**Datasets & Backbone LLMs.** We evaluate our FlyThinker on three long-form personalized generation tasks from the user-based split of the LONGLAMP benchmark (Kumar et al., 2024): *Product Review*, *Abstract Generation*, and *Topic Writing*, which are designed to assess the effectiveness of LLMs in generating personalized, long-text outputs. The model is required to generate a long-form response that aligns with user preferences. Our primary backbone is Qwen2.5-3B-Instruct, with additional experiments on Qwen2.5-7B-Instruct and Gemma-7B-it available in Appendix L and M.

Table 1: Main results on our FlyThinker method across three datasets: *Product Review*, *Abstract Generation*, and *Topic Writing*. FlyThinker consistently outperforms both tuning-free and tuning-based baselines.

| Methods (↓) | Product Review | | | | Abstract Generation | | | | Topic Writing | | | |
|---|---|---|---|---|---|---|---|---|---|---|---|---|
| | ROUGE-1 | ROUGE-L | BLEU | METEOR | ROUGE-1 | ROUGE-L | BLEU | METEOR | ROUGE-1 | ROUGE-L | BLEU | METEOR |
| Non-pers | 0.2839 | 0.1317 | 1.5357 | 0.1783 | 0.3276 | 0.1717 | 4.5814 | 0.2476 | 0.2389 | 0.1057 | 1.1190 | 0.1869 |
| RAG | 0.3176 | 0.1420 | 3.2990 | 0.2309 | 0.3168 | 0.1616 | 3.4047 | 0.2998 | 0.2533 | 0.1099 | 1.4339 | 0.2253 |
| CoS | 0.2781 | 0.1347 | 2.9677 | 0.2189 | 0.2973 | 0.1576 | 3.2591 | 0.3023 | 0.2307 | 0.1056 | 1.5426 | 0.2134 |
| LLM-TRSR | 0.3377 | 0.1450 | 2.3279 | 0.2122 | 0.3583 | 0.2056 | 5.5359 | 0.2371 | 0.2839 | 0.1280 | 1.5552 | 0.1922 |
| NextQuill | 0.3311 | 0.1518 | 2.6599 | 0.2172 | 0.3501 | 0.2038 | 5.4388 | 0.2310 | 0.2755 | 0.1269 | 2.1006 | 0.1817 |
| SFT | 0.3551 | 0.1536 | 3.9112 | 0.2309 | 0.3602 | 0.2062 | 5.8189 | 0.2392 | 0.2916 | 0.1345 | 3.8876 | 0.2013 |
| CoT | 0.3466 | 0.1508 | 3.3705 | 0.2220 | 0.3685 | 0.2071 | 5.8538 | 0.2469 | 0.2846 | 0.1304 | 3.0020 | 0.1967 |
| Coconut | 0.3381 | 0.1496 | 3.3213 | 0.2093 | 0.3542 | 0.2041 | 5.2378 | 0.2298 | 0.2786 | 0.1317 | 3.0736 | 0.1823 |
| **FlyThinker** | **0.3663** | **0.1560** | **4.3620** | **0.2489** | **0.3716** | **0.2090** | **6.3383** | 0.2549 | **0.3139** | **0.1362** | **4.0606** | **0.2409** |

**Baselines & Evaluation Metrics.** We compare FlyThinker against the following baselines: (1) *Tuning-free methods*: query-only generation (**Non-pers**) (Yang et al., 2024), retrieval-augmented generation (**RAG**) (Lei et al., 2023), and context steering (**CoS**) (He et al., 2025); (2) *Tuning-based methods*: supervised fine-tuning (**SFT**) (Salemi et al., 2024), summary-enhanced training (**LLM-TRSR**) (Zheng et al., 2024), causal preference modeling (**NextQuill**) (Zhao et al., 2025c), chain-of-thought reasoning (**CoT**) (Wei et al., 2022; Salemi et al., 2025b), and latent reasoning (**Coconut**) (Hao et al., 2024). We report ROUGE-1, ROUGE-L, BLEU, and METEOR scores. Baseline details and results on other evaluation metrics are provided in the Appendix D and E, respectively.

## 4.2 PERFORMANCE ON PERSONALIZED LONG-FORM GENERATION (RQ1)

The main results are summarized in Table 1. From the table, we draw three major observations:

**Obs 1: Consistent gains over strong baselines.** Across all three tasks, FlyThinker surpasses SFT and other tuning-based methods on most metrics. For example, on Product Review, FlyThinker achieves a ROUGE-1 of 0.3663 (+3.1% over SFT) and a BLEU of 4.36 (+11.5%). Similar trends hold for Abstract Generation and Topic Writing, where FlyThinker improves BLEU by about 10% over SFT and delivers higher ROUGE-L scores. These results demonstrate that FlyThinker captures more salient user-specific information and produces outputs that are both lexically and structurally closer to the references.

**Obs 2: Robustness across domains.** FlyThinker maintains strong performance regardless of task type or text length. Its improvements are especially pronounced on Abstract Generation, where it reaches a BLEU of 6.34, outperforming CoT (5.85 BLEU) and Coconut (5.23 BLEU). This suggests that the think-while-generating paradigm is particularly effective in preserving coherence and relevance over long sequences, adapting dynamically as content evolves.

**Obs 3: Superior to latent-reasoning baselines.** Compared with Coconut, which employs a think-then-generate latent reasoning strategy, FlyThinker consistently yields higher ROUGE and BLEU across all tasks. This confirms that our interleaving reasoning with generation enables more fine-grained control and better exploitation of user context than static latent reasoning approach.

## 4.3 TRAINING AND INFERENCE EFFICIENCY OF FLYTHINKER (RQ2)

We further analyze the efficiency of FlyThinker. We report both training time per epoch and inference latency with fixed output length for fair comparison (Figure 3).

**Obs 1: In terms of training efficiency, FlyThinker trains much faster than other reasoning-based baselines while remaining close to SFT.** Although FlyThinker produces latent reasoning for every token position, its parallel reasoning mechanism adds only a minor overhead relative to SFT. Crucially, this overhead remains far lower than CoT and Coconut, whose autoregressive reasoning requires sequential token generation and substantially slows down training. When the Reasoner is scaled down, FlyThinker's cost approaches that of plain SFT (see Section 3.3), demonstrating that parallel reasoning generation effectively amortizes the additional computation.

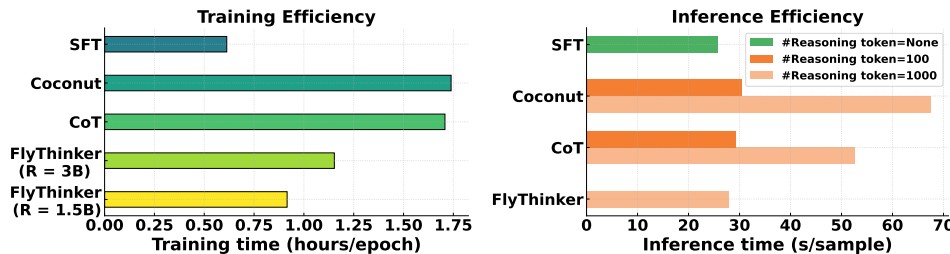

Figure 3: Training and inference efficiency of FlyThinker. The formal results are detailed in the Appendix N.

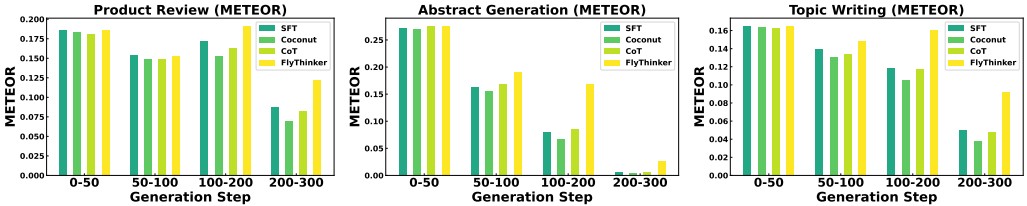

Figure 4: Position-sensitive evaluation across three datasets. Each generated sequence is divided into four segments by token position. We report METEOR here, with results on other metrics provided in Appendix F.

**Obs 2: In terms of inference efficiency, FlyThinker delivers much faster inference than reasoning-based methods and nearly matches SFT latency.** During decoding, FlyThinker achieves near-SFT latency by paralleling reasoning and token prediction: while the Generator predicts the next token, the Reasoner simultaneously prepares reasoning for the next step. This staggered, parallel design removes the sequential bottleneck inherent in CoT and Coconut. As a result, FlyThinker delivers token-level reasoning with negligible latency overhead.

### 4.4 POSITION-SENSITIVE EVALUATION OF FLYTHINKER (RQ3)

To better understand FlyThinker's impact on personalized long-form generation, we analyze how generation quality evolves across token positions. We segment each generated sequence into contiguous chunks based on token order and compute personalization quality for each segment, enabling a fine-grained view of how performance changes as the output grows longer. Figure 4 reports the results on the METEOR metric, while results on other metrics are provided in the Appendix F.

**Obs 1: Personalized generation quality degrades in later segments for all baselines.** As shown in Figure 4, methods such as SFT, CoT, and Coconut exhibit a clear downward trend in ROUGE and BLEU scores as token positions increase, reflecting the well-known "context drift" problem in long-form generation: models gradually lose alignment with user preferences as the text becomes longer.

**Obs 2: FlyThinker effectively mitigates quality degradation and sustains personalization.** In the [100, 200] and [200, 300] token ranges, FlyThinker consistently outperforms all baselines by a substantial margin, demonstrating its ability to preserve personalization quality even in the most challenging later segments. We attribute this improvement to FlyThinker's on-the-fly latent reasoning, which performs step-wise context-aware reasoning and continuously refreshes personalized cues from earlier context. This adaptive mechanism provides stronger guidance for later token predictions, mitigating context drift and yielding more faithful, user-aligned outputs.

### 4.5 ABLATION STUDY (RQ4)

We conduct a comprehensive ablation study to examine the design choices of FlyThinker, focusing on three factors: the scale of the Reasoner, the effect of the weighting parameter $\lambda$. An additional study about the position to add latent reasoning tokens is provided in Appendix H.

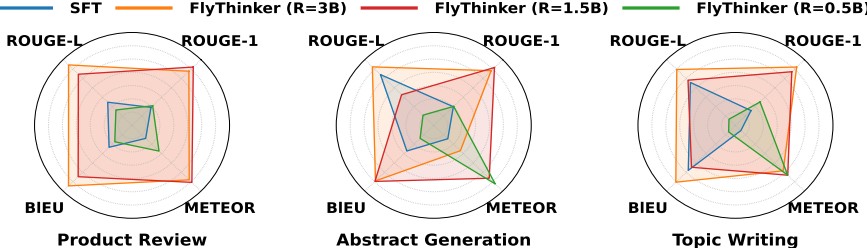

Figure 5: Ablation study on Reasoner scale. We compare our FlyThinker with different Reasoner sizes (3B, 1.5B, 0.5B) on three datasets. The full experimental results are provided in the table of Appendix O.

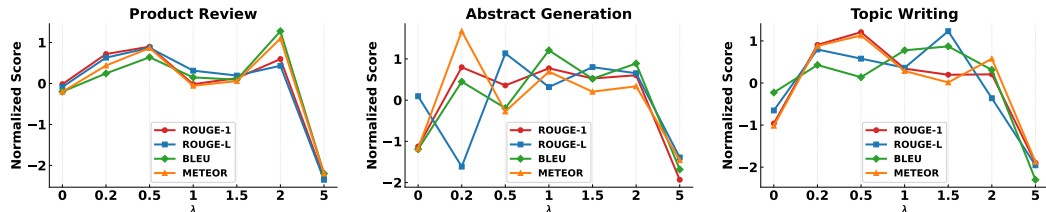

Figure 6: Sensitivity analysis of the weighting parameter $\lambda$ on three datasets. Moderate values (0.5–2) yield the best overall performance. The full experimental results are provided in the table of Appendix P.

**(1) Analysis of Reasoner Scale.** Section 4.3 showed that reducing the size of the Reasoner significantly improves training efficiency. Here, we investigate whether such downsizing impacts generation quality. We evaluate FlyThinker using Qwen2.5-3B-Instruct, Qwen2.5-1.5B-Instruct, and Qwen2.5-0.5B-Instruct as the Reasoner backbones, and report results in Figure 5. Two key observations emerge:

*Obs 1: Moderate downsizing preserves generation quality.* When reducing the latent reasoner size from 3B to 1.5B, FlyThinker achieves nearly identical performance across ROUGE-1/L, BLEU, and METEOR. This demonstrates that significant efficiency gains can be achieved without sacrificing output quality, indicating a favorable cost–performance trade-off.

*Obs 2: Aggressive downsizing leads to performance degradation.* Using a 0.5B Reasoner results in clear drops in ROUGE-L and BLEU, suggesting that overly small Reasoners lack the capacity to provide sufficiently rich reasoning signals for high-quality generation.

**(2) Analysis for the weighting parameter $\lambda$.** We analyze the sensitivity of FlyThinker to the hyperparameter $\lambda$ in Eq. 6, which controls the contribution of latent reasoning. We sweep $\lambda \in \{0, 0.2, 0.5, 1, 1.5, 2, 5\}$ and report normalized results across all metrics in Figure 6. We make the following observations: 1) a moderate weighting strikes the best balance between leveraging reasoning signals and preserving stable generation. However, extreme $\lambda$ values degrade performance. Very small $\lambda$ suppresses the influence of latent reasoning, underutilizing reasoning capacity, whereas overly large $\lambda$ destabilizes the reasoning process and lowers overall quality. 2) When $\lambda$ varies within [0.2, 2.0], individual metrics fluctuate slightly, but all results remain above the SFT baseline, indicating that our method's effectiveness is relatively stable within this range.

## 5 RELATED WORK

**Personalized Long-Form Generation.** As the demand for complex content—such as customized reports, creative narratives, and personalized analyses—continues to grow, personalized long-form generation has emerged as a critical research frontier (Salemi et al., 2025b; Kumar et al., 2024; Salemi et al., 2025a; Salemi & Zamani, 2025). Unlike short-text generation, long-form tasks require LLMs to maintain consistency and personalization quality over extended sequences. This requirement is often challenged by the *context drift* problem, in which LLMs gradually lose alignment with user preferences as the text length increases (Kumar et al., 2024; Salemi et al., 2025b).

The recent benchmark LongLaMP (Kumar et al., 2024) has been proposed to systematically evaluate LLMs' capabilities in personalized long-form generation. To address quality degradation, some approaches incorporate Retrieval-Augmented Generation (RAG), which injects personalized signals into the LLMs' context (Mysore et al., 2023a; Kumar et al., 2024). REST-PG to improve the LLM's ability to leverage personal data more effectively throughout the generation process (Salemi et al., 2025b).

Despite these advances, a critical gap remains: existing methods struggle to reason over implicit user preferences and dynamically adapt to evolving contexts in complex, real-world long-form scenarios. FlyThinker directly addresses this limitation by introducing a dynamic, token-level reasoning mechanism that continuously integrates personalized signals during generation. By enabling fine-grained, adaptive personalization rather than relying on static conditioning strategies, FlyThinker achieves deeper contextual alignment and more robust personalization across extended outputs.

**Reasoning-Enhanced Generation.** Reasoning-enhanced generation leverages the reasoning capabilities of LLMs to guide the production of high-quality responses. Widely adopted paradigms such as explicit reasoning (Wei et al., 2022) and latent reasoning (Hao et al., 2024) encourage LLMs to articulate intermediate reasoning steps. These reasoning paradigms have been adapted for personalization to capture implicit user preferences. REST-PG (Salemi et al., 2025b) employs a think-then-generate paradigm in which reasoning paths over personal context are generated via self-training to improve alignment. R2P (Luo et al., 2025a) utilizes a hierarchical reasoning template to guide large reasoning models in a think-then-generate framework for structured personalized outputs. RPM (Kim et al., 2025b) introduces personalization for black-box LLMs by constructing structured, user-specific reasoning paths and retrieving reasoning-aligned examples prior to producing the final response. TagPR (Jin et al., 2025) enhances personalization reasoning via semantically tagged reasoning chains and reinforcement learning over a personalization reward model. AlignXplore (Li et al., 2025a) infers personalized preferences from behavioral signals through extended inductive reasoning chains before downstream generation.

The aforementioned methods primarily rely on a static, one-shot reasoning process, in which the LLM performs reasoning once before generating the entire response. While effective for short-form tasks (Luo et al., 2025a; Kim et al., 2025b; Jin et al., 2025; Li et al., 2025a), this think-then-generate paradigm often struggles with long-form personalized generation because it fails to adapt to evolving contexts and dynamic preference shifts as the content unfolds. Furthermore, the sequential separation of reasoning and generation introduces significant latency, limiting real-time applicability (Salemi et al., 2025b).

In contrast, by employing a parallel architecture that synchronizes a reasoning model with a generation model, our proposed FlyThinker enables on-the-fly reasoning that evolves dynamically alongside the response. This approach not only mitigates the training and inference efficiency bottlenecks inherent in sequential paradigms but also significantly enhances the LLM's capacity for fine-grained, dynamic reasoning over user preferences, resulting in superior personalization performance in complex long-form tasks.

## 6 CONCLUSION

In this work, we introduced FlyThinker, an efficient framework for LLM personalization built on a think-while-generating paradigm. Unlike prior methods that rely on static reasoning, FlyThinker dynamically integrates token-level latent reasoning into the generation process, enabling models to better capture evolving user preferences during long-form text creation. The framework maintains parallelism in both training and inference, addressing the efficiency bottlenecks inherent in traditional reasoning-augmented generation. Experiments show clear gains in both personalization quality and efficiency over strong baselines.

## ETHICS STATEMENT

This work complies with the ICLR Code of Ethics. No human or animal subjects were involved. All datasets (Product Review, Abstract Generation, Topic Writing) were used in accordance with

their usage guidelines, without privacy violations or personally identifiable information. We are committed to maintaining transparency, fairness, and integrity in all aspects of this research.

## REPRODUCIBILITY STATEMENT

We have made every effort to ensure the reproducibility of our results. The experimental setup is detailed in Section 4.1 and Appendix C. We also provide pseudocode for both training and inference (see Appendix B) to assist in reproducing our experiments. Moreover, the datasets in LongLamp are publicly available, ensuring consistent and reproducible evaluation. We believe these measures will enable other researchers to replicate our work and advance the field.

## ACKNOWLEDGMENTS

This work is supported by the National Natural Science Foundation of China (62525211) and the advanced computing resources provided by the Supercomputing Center of the USTC.

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

## A  LLM USAGE

Large Language Models (LLMs) were used to aid in the writing and polishing of the manuscript. Specifically, we used an LLM to assist in sentence rephrasing, grammar checking, and enhancing readability and flow. The LLM was not involved in the ideation, methodology, or data analysis. All research concepts, ideas, and analyses were developed and conducted by the authors.

The authors take full responsibility for the manuscript's content, including any text generated or polished by the LLM. All use complied with ethical standards, ensuring no plagiarism or scientific misconduct.

## B  ALGORITHM PSEUDOCODE

---

**Algorithm 1** Training FlyThinker

**Require:** Generator $G_\phi$; Reasoner $R_\theta$; token embedding $e(\cdot)$; fusion coef. $\lambda$; batch sample sequences $x[:,0:T]$
1: **(One-pass latent thoughts)**
2: $r[:,0:T] \leftarrow R_\theta(x[:,0:T])$
3: **(Build enhanced embeddings)**
4: $e[:,1:T] \leftarrow e(x[:,1:T]) + \lambda \cdot r[:,0:T-1]$
5: **(Computing loss)**
6: $P_t \leftarrow G_\phi(e[:,0:T])$
7: $\mathcal{L} \leftarrow \mathrm{CE}(P_t, y_t)$
8: Update $\theta, \phi$ by backprop on $\mathcal{L}$
9: **return** trained $R_\theta, G_\phi$

---

**Algorithm 2** Inference with FlyThinker

**Require:** Trained $G_\phi, R_\theta$; max steps $M$; token list $s$; $x_0, e_0$ is derived from the context in the same way as during training.
1: **for** step $t = 1, 2, \ldots, M$ **do**
2:    **(Parallel)** $r_{t-1}^{(L)} \leftarrow R_\theta(x_{<t-1}, x_{t-1})$
3:    **(Parallel)** $P_t \leftarrow G_\phi(e_{<t-1}, e_{t-1})$
4:    Sample $y_t \sim P_t$, $s \leftarrow s + y_t$
5:    **(Prepare for next step)**
6:    $x_t \leftarrow y_t$
7:    $e_t \leftarrow e(x_{t-1}) + \lambda \cdot r_{t-1}^{(L)}$
8: **end for**
9: **return** full generation sequence $s$

---

## C  IMPLEMENTATION DETAILS

In our study, we fine-tune the Qwen2.5-3B-Instruct and Qwen2.5-7B-Instruct model to validate our approach. All experiments, including FlyThinker and baseline models, are conducted on four NVIDIA A100 GPUs. The LLMs are loaded via HuggingFace Transformers (Wolf et al., 2019), and the training is implemented in Python 3.10 using the Verl training framework (Sheng et al., 2024).

## D  DETAILS OF BASELINE METHODS

We provide a more detailed description of the baselines considered in our comparisons:

**Non-pers**: Use the original LLM and the user query directly, without any personalization alignment, as the default generation method.

**RAG**: Supply the original LLM with user-specific information retrieved from interaction history and inject it into the input context, enabling personalization without training.

**CoS**: Compute contextual influence by contrasting output probabilities with and without context, and adjust it with a parameter $\lambda$ to control personalization strength.

**SFT**: Fine-tune the LLM with user interaction history, queries, and ground-truth responses, aligning parameters toward personalized outputs.

**LLM-TRSR**: Extract key information from the user's interaction history, integrate the resulting summary into prompts, and fine-tune the LLM to improve personalization.

**NextQuill**: Fine-tune the LLM by modeling causal preference effects of user history on token prediction, and align the LLM's internal preferences with those observed in real data, focusing on preference-bearing tokens.

**CoT**: Fine-tune the LLM with explicit reasoning paths under a "think-then-generate" paradigm, generating reasoning steps before producing the final answer.

**Coconut**: Fine-tune the LLM by generating latent representations autoregressively as an latent reasoning path, and produce the final answer conditioned on them.

## E    COMPREHENSIVE EVALUATION RESULTS

This section further incorporates the BERTScore metric and the average relative improvement, providing a more comprehensive demonstration of FlyThinker's advantages. The results are presented in Table 12.

## F    POSITION-SENSITIVE EVALUATION

This section presents additional results on position-sensitive evaluation (see Section 4.4). As illustrated in Figure 8, we can observe that FlyThinker effectively alleviates quality degradation and consistently preserves personalization across all datasets and metrics.

## G    LATENT REASONING TRAJECTORY: VISUALIZATION AND ANALYSIS

Interpretablity is common challenge for latent reasoning. Draw inspiration from your comments, we have conducted an visualization study. Under a fixed task, we selected five users and extracted one sample from each of them. We compared the Reasoner latent reasoning tokens for these samples and visualized them using t-SNE. We then used linear regression to estimate the direction from early to later reasoning tokens, representing each user's reasoning trajectory. The corresponding Figure 7 for this experiment illustrates that different users show different trajectory directions.

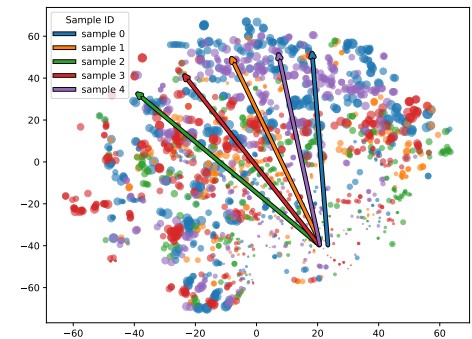

Figure 7: t-SNE visualization and trajectory regression vectors of latent reasoning tokens under two experimental configurations.

To examine whether these trajectories relate to the preference, we used GPT-5.1 to compare the samples. We selected the two samples whose users had the most similar trajectory directions (sample0 and sample4) and the one farthest away (sample2) in Figure 7. GPT-5.1 judged sample4 to be most similar to sample0 in paragraph structure, narrative style, and analytical approach. Sample2 was the least similar. These results show that different users induce distinct reasoning trajectories, and that closer trajectories correspond to more similar writing styles and semantic structures, etc. This suggests that latent reasoning tokens may contain interpretable semantic information useful for user clustering and preference visualization.

## H    EFFECT OF THE POSITION OF LATENT REASONING AUGMENTATION.

When implementing FlyThinker, we also add latent reasoning tokens to the task prompt and query parts of the model inputs to ensure alignment of the representation space across the entire sequence. We further study where to apply latent reasoning augmentation: only on the input tokens (*Input-only*), only on the output tokens (*Output- only*), or jointly on both as in *FlyThinker (global enhancement)*. Results are shown in Table 2.

*Obs 1: Global enhancement yields the most consistent improvements.* FlyThinker, which applies reasoning augmentation at both input and output positions, achieves the highest overall performance across tasks and metrics, with especially notable gains in ROUGE and BLEU. This indicates that combining input- and output-side reasoning provides complementary benefits, resulting in more faithful and informative generation.

*Obs 2: Input-only enhancement increases diversity but harms alignment.* Injecting reasoning solely at the input stage makes the model more "free-form," producing diverse paraphrases but reducing lexical and structural alignment with reference texts, which leads to lower ROUGE/BLEU scores.

*Obs 3: Output-only enhancement underperforms due to weak contextual grounding.* Reasoning injected exclusively at the output stage lacks sufficient conditioning on the user profile and query, resulting in consistently weaker performance across tasks.

| Methods ($\downarrow$) | Product Review | | | | Abstract Generation | | | | Topic Writing | | | |
|---|---|---|---|---|---|---|---|---|---|---|---|---|
| | ROUGE-1 | ROUGE-L | BLEU | METEOR | ROUGE-1 | ROUGE-L | BLEU | METEOR | ROUGE-1 | ROUGE-L | BLEU | METEOR |
| **SFT** | 0.3551 | 0.1536 | 3.9112 | 0.2309 | 0.3602 | 0.2062 | 5.8189 | 0.2392 | 0.2916 | 0.1345 | 3.8876 | 0.2013 |
| **Input-only** | 0.3559 | 0.1471 | 4.0807 | **0.2899** | 0.3357 | 0.1759 | 4.5466 | **0.3036** | 0.2783 | 0.1195 | 2.7058 | **0.2651** |
| **Output-only** | 0.3417 | 0.1513 | 3.3729 | 0.2123 | 0.3586 | 0.2064 | 5.7380 | 0.2355 | 0.2740 | 0.1314 | 2.9972 | 0.1759 |
| **FlyThinker** | **0.3663** | **0.1560** | **4.3620** | 0.2489 | **0.3716** | **0.2090** | **6.3383** | 0.2549 | **0.3139** | **0.1362** | **4.0606** | 0.2409 |

Table 2: Ablation study on the position of latent reasoning augmentation. We compare our Fly-Thinker with reasoning applied only to input tokens (**Input-only**) or only to output tokens (**Output-only**).

## I    REASONER–GENERATOR RELATIVE SIZE ANALYSIS IN FLYTHINKER

Table 3: Evaluation of FlyThinker under different Reasoner–Generator size pairings, comparing 0.5B/1.5B Reasoners with 3B/7B Generators.

| | G = 3B | | | | | G = 7B | | | | |
|---|---|---|---|---|---|---|---|---|---|---|
| **Product Review** | ROUGE-1 | ROUGE-L | BLEU | METEOR | BERT-SCORE | ROUGE-1 | ROUGE-L | BLEU | METEOR | BERT-SCORE |
| SFT | 0.3551 | 0.1536 | 3.9112 | 0.2309 | 0.7681 | 0.3512 | 0.1560 | 3.8953 | 0.2224 | 0.7674 |
| FlyThinker (R=1.5B) | **0.3676** | **0.1554** | **4.2123** | **0.2501** | **0.7729** | 0.3738 | 0.1578 | 5.1072 | 0.2715 | **0.7846** |
| FlyThinker (R=0.5B) | 0.3556 | 0.1531 | 3.8072 | 0.2366 | 0.7692 | 0.3712 | 0.1566 | **4.8035** | **0.2737** | 0.7813 |
| **Abstract Generation** | ROUGE-1 | ROUGE-L | BLEU | METEOR | BERT-SCORE | ROUGE-1 | ROUGE-L | BLEU | METEOR | BERT-SCORE |
| SFT | 0.3602 | **0.2062** | 5.8189 | 0.2392 | 0.8812 | 0.3645 | **0.2122** | 5.9661 | 0.2398 | 0.8854 |
| FlyThinker (R=1.5B) | **0.3726** | 0.1989 | **6.2690** | 0.2910 | **0.8896** | 0.3708 | 0.1971 | **6.2732** | 0.3010 | **0.8910** |
| FlyThinker (R=0.5B) | 0.3603 | 0.1915 | 5.5334 | **0.2985** | 0.8883 | 0.3665 | 0.1949 | 5.9182 | **0.3012** | 0.8903 |
| **Topic Writing** | ROUGE-1 | ROUGE-L | BLEU | METEOR | BERT-SCORE | ROUGE-1 | ROUGE-L | BLEU | METEOR | BERT-SCORE |
| SFT | 0.2916 | 0.1345 | **3.8876** | 0.2013 | 0.7067 | 0.2849 | **0.1351** | 3.4850 | 0.1883 | 0.7071 |
| FlyThinker (R=1.5B) | **0.3115** | **0.1348** | 3.8004 | 0.2450 | 0.7154 | **0.3076** | 0.1325 | **3.6572** | 0.2564 | **0.7249** |
| FlyThinker (R=0.5B) | 0.2960 | 0.1297 | 3.2908 | **0.2450** | **0.7158** | 0.3023 | 0.1319 | 3.4868 | **0.2566** | 0.7215 |

Understanding the relative sizing between the Reasoner and the Generator is essential for deploying FlyThinker effectively in practical scenarios. In this section, we conducted a systematic analysis using 3B and 7B Generators, and tested Reasoners with sizes of 0.5B and 1.5B. As shown in Table 3, with a 7B Generator, a 0.5B Reasoner can already deliver performance improvements, while with a 3B Generator, the Reasoner needs to reach 1.5B to achieve meaningful and consistent gains across metrics. These results suggest that a generally small Reasoner (around 1.5B) can be effective. However, there is no clear proportional size requirement between the Reasoner and the Generator — a stronger Generator places fewer demands on the Reasoner. Naturally, a larger Reasoner typically delivers better performance.

## J    ROBUSTNESS OF FLYTHINKER UNDER VARYING USER HISTORY LENGTHS

In this section, we analyzed FlyThinker's performance under different history lengths. Specifically, we tested three settings: two history examples ($\approx 1,750$ tokens), three history examples ($\approx 2,250$ tokens), and four history examples ($\approx 2,750$ tokens), comparing our method with the SFT method. The results, summarized in the Table 4, show that performance increases as the context length grows across most metrics, while FlyThinker consistently outperforms SFT in all tested settings. These findings suggest that our method is relatively insensitive to user history length.

## K    LLM-BASED EVALUATION OF FLYTHINKER

In this section, we added an LLM-based evaluation to provide a more comprehensive assessment of personalized generation quality. Specifically, we use GPT-4o as an evaluator and compare the

Table 4: Robustness of FlyThinker to different user history lengths. Increasing the number of history examples yields mild performance gains while maintaining a clear advantage over SFT.

| | [#his=2, #his_token≈1750] | | [#his=3, #his_token≈2250] | | [#his=4, #his_token≈2750] | |
| --- | --- | --- | --- | --- | --- | --- |
| **Product Review** | **SFT** | **FlyThinker** | **SFT** | **FlyThinker** | **SFT** | **FlyThinker** |
| ROUGE-1 | 0.3551 | 0.3663 | 0.3619 | 0.3705 | 0.3635 | 0.3719 |
| ROUGE-L | 0.1536 | 0.1560 | 0.1568 | 0.1581 | 0.1585 | 0.1619 |
| BLEU | 3.9112 | 4.3620 | 4.4810 | 4.5020 | 5.1703 | 5.0671 |
| METEOR | 0.2309 | 0.2489 | 0.2394 | 0.2490 | 0.2437 | 0.2511 |
| BERT-SCORE | 0.7681 | 0.7755 | 0.7711 | 0.7762 | 0.7708 | 0.7772 |
| **Abstract Generation** | **SFT** | **FlyThinker** | **SFT** | **FlyThinker** | **SFT** | **FlyThinker** |
| ROUGE-1 | 0.3602 | 0.3716 | 0.3648 | 0.3786 | 0.3651 | 0.3795 |
| ROUGE-L | 0.2062 | 0.2090 | 0.2095 | 0.2035 | 0.2102 | 0.2047 |
| BLEU | 5.8189 | 6.3383 | 5.9950 | 6.8119 | 6.0234 | 7.0848 |
| METEOR | 0.2392 | 0.2549 | 0.2422 | 0.2903 | 0.2432 | 0.2882 |
| BERT-SCORE | 0.8812 | 0.8858 | 0.8845 | 0.891 | 0.8852 | 0.891 |
| **Topic Writing** | **SFT** | **FlyThinker** | **SFT** | **FlyThinker** | **SFT** | **FlyThinker** |
| ROUGE-1 | 0.2916 | 0.3139 | 0.2963 | 0.3153 | 0.298 | 0.3185 |
| ROUGE-L | 0.1345 | 0.1362 | 0.1360 | 0.1376 | 0.1359 | 0.1401 |
| BLEU | 3.8876 | 4.0606 | 4.8021 | 4.4026 | 4.7817 | 4.8841 |
| METEOR | 0.2013 | 0.2409 | 0.2084 | 0.2384 | 0.2126 | 0.2395 |
| BERT-SCORE | 0.7067 | 0.7205 | 0.7092 | 0.7197 | 0.7091 | 0.7204 |

outputs produced by FlyThinker with those from the strongest baseline, SFT, following a structured evaluation rubric 9. As shown in the Table 5, FlyThinker consistently achieves higher scores across all datasets under the LLM-based evaluation. These findings indicate that FlyThinker's personalization gains are reflected not only in traditional automatic metrics but also in the LLM judge's assessments.

Table 5: LLM-Based Evaluation of FlyThinker.

| | GPT-4o-mini evaluation | | |
| --- | --- | --- | --- |
| | **Product Review** | **Abstract Generation** | **Topic Writing** |
| SFT | 0.3695 | 0.5718 | 0.5718 |
| FlyThinker | **0.3816** | **0.5955** | **0.5955** |

## L GENERATOR SCALING STUDY

We conduct a study on generator scaling with Qwen2.5-7B-Instruct , aiming to evaluate scalability. For this purpose, we adopt a simple yet strong baseline—SFT—as our reference baseline. As shown in Table 6, FlyThinker-7B (R=1.5B) consistently achieves better overall performance across the three tasks. In particular, FlyThinker demonstrates clear advantages on BLEU,METEOR, indicating improved lexical precision and semantic alignment. Although SFT-7B remains competitive and sometimes stronger on ROUGE-L (e.g., in abstract generation and topic writing), FlyThinker provides more balanced improvements across multiple evaluation metrics. These results highlight the effectiveness and robustness of FlyThinker.

## M CROSS-ARCHITECTURE VALIDATION ON ADDITIONAL MODELS

On the Gemma-7B-it model, FlyThinker also shows consistent performance gains (ref. Table 7). Together with the results on Qwen2.5-3B and 7B, these findings indicate that our approach demonstrates strong generalization across different model architectures.

Table 6: The evaluation results on Qwen2.5-7B-Instruct. The results show that FlyThinker consistently outperforms the SFT baseline across all datasets.

| Qwen2.5-7B-Instruct | | | | | |
|---|---|---|---|---|---|
| **Product Review** | ROUGE-1 | ROUGE-L | BLEU | METEOR | BERT-SCORE |
| SFT | 0.3512 | 0.1560 | 3.8953 | 0.2224 | 0.7674 |
| FlyThinker (R=1.5B) | **0.3738** | **0.1578** | **5.1072** | **0.2715** | **0.7846** |
| **Abstract Generation** | ROUGE-1 | ROUGE-L | BLEU | METEOR | BERT-SCORE |
| SFT | 0.3645 | **0.2122** | 5.9661 | 0.2398 | 0.8854 |
| FlyThinker (R=1.5B) | **0.3708** | 0.1971 | **6.2732** | **0.3010** | **0.8910** |
| **Topic Writing** | ROUGE-1 | ROUGE-L | BLEU | METEOR | BERT-SCORE |
| SFT | 0.2849 | **0.1351** | 3.4850 | 0.1883 | 0.7071 |
| FlyThinker (R=1.5B) | **0.3076** | 0.1325 | **3.6572** | **0.2564** | **0.7249** |

Table 7: Cross-architecture evaluation on Gemma-7B-it, demonstrating that FlyThinker achieves stable performance gains over SFT, confirming its effectiveness across different model families.

| Gemma-7B-it | | | | | |
|---|---|---|---|---|---|
| **Product Review** | ROUGE-1 | ROUGE-L | BLEU | METEOR | BERT-SCORE |
| SFT | 0.3285 | 0.1544 | **3.9774** | **0.2166** | **0.7559** |
| FlyThinker (R=2B) | **0.3355** | **0.1579** | 3.5646 | 0.2160 | 0.7553 |
| **Abstract Generation** | ROUGE-1 | ROUGE-L | BLEU | METEOR | BERT-SCORE |
| SFT | 0.3501 | 0.2057 | 6.1330 | 0.2404 | 0.8750 |
| FlyThinker (R=2B) | **0.3624** | **0.2062** | **6.8307** | **0.2638** | **0.8800** |
| **Topic Writing** | ROUGE-1 | ROUGE-L | BLEU | METEOR | BERT-SCORE |
| SFT | 0.2664 | 0.1323 | **3.9175** | 0.1938 | 0.6960 |
| FlyThinker (R=2B) | **0.2829** | **0.1385** | 3.6046 | **0.2035** | **0.6989** |

# N    MEMORY AND COMPUTATION ANALYSIS OF FLYTHINKER

In this section, we have included a more comprehensive evaluation of memory, computational, and time-latency costs, summarized in the Table 8. The symbols are defined as:

($\theta_G$ / $\theta_R$): parameters of the generator and reasoner, respectively

($N_1$ / $N_2$): number of text tokens / reasoning tokens to generate

($\mathcal{C}_G$ / $\mathcal{C}_R$): cost of one forward pass for the generator / reasoner

($\ell_G$ / $\ell_R$): latency to generate one token by the generator / reasoner

Specifically, regarding computational cost you focused: our method's cost is approximately ($N_1\mathcal{C}_G + N_2\mathcal{C}_R$), while the baseline latent reasoning method incurs ($N_1\mathcal{C}_G + N_2\mathcal{C}_G$). The difference comes from ($N_2\mathcal{C}_R$) versus ($N_2\mathcal{C}_G$). Since the reasoner is typically much smaller than the generator (e.g., 0.5B vs. 7B), our method's computational cost is significantly lower than latent reasoning.

Compared to the SFT baseline (cost ($N_1\mathcal{C}_G$)), our additional cost is ($N_2\mathcal{C}_R$). As ($\mathcal{C}_R \ll \mathcal{C}_G$), this overhead is relatively small. For instance, using a 0.5B reasoner for a 7B generator with ($N_1 = N_2 = 1000$), the FLOPS are: our method — 15 TFLOPs, Cocount—28 TFLOPs, SFT — 14 TFLOPs. In this scenario, our method adds only 1TFLOPs computation cost while delivering significant performance improvements, compared to SFT.

Importantly, a key advantage of our method is reduced latency, as also shown in the Table 8.

Table 8: Analysis of FlyThinker's memory and computation characteristics, comparing its training and inference overhead with SFT and other reasoning-based baselines.

| Method | Memory | Computation | Latency | |
|---|---|---|---|---|
| | | | Training stage | Inference Stage |
| SFT | $|\theta_G|$ | $N_1\mathcal{C}_G$ | $\ell_G$ | $N_1\ell_G$ |
| CoT | $|\theta_G|$ | $N_1\mathcal{C}_G + N_2\mathcal{C}_G$ | $\ell_G + N_2\ell_G$ | $N_1\ell_G + N_2\ell_G$ |
| Coconut | $|\theta_G|$ | $N_1\mathcal{C}_G + N_2\mathcal{C}_G$ | $\ell_G + N_2\ell_G$ | $N_1\ell_G + N_2\ell_G$ |
| FlyThinker | $|\theta_G| + |\theta_R|$ | $N_1\mathcal{C}_G + N_2\mathcal{C}_R$ | $\ell_G + \ell_R$ | $\approx N_1\ell_G$ |

## O    FULL EXPERIMENTAL RESULTS ON REASONER SCALE IN FIGURE 5

In Figure 5, all metrics were normalized and plotted on a unified scale to highlight relative differences across methods. In this section, we supplement Figure 5 with the complete numerical results (see Table 9).

Table 9: Complete evaluation results in Figure 5, providing absolute metric values to supplement the normalized plot in the main text.

| Product Review | ROUGE-1 | ROUGE-L | BLEU | METEOR | BERT-SCORE |
|---|---|---|---|---|---|
| SFT | 0.3551 | 0.1536 | 3.9112 | 0.2309 | 0.7681 |
| FlyThinker (R=3B) | 0.3663 | **0.1560** | **4.3620** | 0.2489 | **0.7755** |
| FlyThinker (R=1.5B) | **0.3676** | 0.1554 | 4.2123 | **0.2501** | 0.7729 |
| FlyThinker (R=0.5B) | 0.3556 | 0.1531 | 3.8072 | 0.2366 | 0.7692 |
| **Abstract Generation** | **ROUGE-1** | **ROUGE-L** | **BLEU** | **METEOR** | **BERT-SCORE** |
| SFT | 0.3602 | 0.2062 | 5.8189 | 0.2392 | 0.8812 |
| FlyThinker (R=3B) | 0.3716 | **0.2090** | **6.3383** | 0.2549 | 0.8858 |
| FlyThinker (R=1.5B) | **0.3726** | 0.1989 | 6.2690 | 0.2910 | **0.8896** |
| FlyThinker (R=0.5B) | 0.3603 | 0.1915 | 5.5334 | **0.2985** | 0.8883 |
| **Topic Writing** | **ROUGE-1** | **ROUGE-L** | **BLEU** | **METEOR** | **BERT-SCORE** |
| SFT | 0.2916 | 0.1345 | 3.8876 | 0.2013 | 0.7067 |
| FlyThinker (R=3B) | **0.3139** | **0.1362** | **4.0606** | 0.2409 | **0.7205** |
| FlyThinker (R=1.5B) | 0.3115 | 0.1348 | 3.8004 | **0.2450** | 0.7154 |
| FlyThinker (R=0.5B) | 0.2960 | 0.1297 | 3.2908 | **0.2450** | 0.7158 |

## P    EXPANDED SENSITIVITY ANALYSIS OF THE HYPERPARAMETER $\lambda$

In this section, we report all raw (unnormalized) metrics in the Table 10 for a clear examination of the effect of the $\lambda$ hyperparameter in the original Figure 6. These extended experiments reveal several insights:

**Fine-grained tuning of $\lambda$ can further improve specific metrics.** For example, in the Abstract Generation task, $\lambda = 1.0$ leads to additional improvements in BLEU, and in the Topic Writing task, $\lambda = 1.5$ further boosts ROUGE-L and BLEU compared with coarser settings.

**Moderate $\lambda$ values provide robust overall performance.** The table shows that when $\lambda$ stays within a moderate range around 1.0—neither too close to 0 nor excessively large (e.g., approaching 5)—the model consistently outperforms the SFT baseline. This suggests that $\lambda$ does not require precise tuning in practice and the method is robust to reasonable variations.

**The performance fluctuations between $\lambda = 0.2$ and 2.0 are mild and acceptable.** When $\lambda$ varies within [0.5, 2.0], individual metrics fluctuate slightly, but all results remain above the SFT baseline, indicating that our method's effectiveness is relatively stable within this range.

Table 10: Complete evaluation results in Figure 10, providing absolute metric values to supplement sensitivity analysis of the $\lambda$ hyperparameter.

| | | | | $\lambda$ | | | |
|---|---|---|---|---|---|---|---|
| **Product Review** | **0** | **0.2** | **0.5** | **1** | **1.5** | **2** | **5** |
| ROUGE-1 | 0.3551 | 0.3641 | **0.3664** | 0.3551 | 0.3569 | 0.3626 | 0.3271 |
| ROUGE-L | 0.1536 | 0.1554 | **0.1560** | 0.1546 | 0.1543 | 0.1550 | 0.1479 |
| BLEU | 3.9112 | 4.1088 | 4.3223 | 4.0979 | 4.0685 | **4.6599** | 2.7995 |
| METEOR | 0.2309 | 0.2419 | 0.2489 | 0.2334 | 0.2353 | **0.2529** | 0.1976 |
| BERT-SCORE | 0.7681 | 0.7723 | **0.7755** | 0.7687 | 0.7688 | 0.7740 | 0.7541 |
| **Abstract Generation** | **0** | **0.2** | **0.5** | **1** | **1.5** | **2** | **5** |
| ROUGE-1 | 0.3602 | **0.3750** | 0.3716 | 0.3748 | 0.3729 | 0.3734 | 0.3540 |
| ROUGE-L | 0.2062 | 0.2016 | **0.2090** | 0.2068 | 0.2081 | 0.2077 | 0.2022 |
| BLEU | 5.8189 | 6.5881 | 6.3383 | **7.0687** | 6.7091 | 6.8036 | 5.4578 |
| METEOR | 0.2392 | **0.2891** | 0.2549 | 0.2718 | 0.2633 | 0.2657 | 0.2345 |
| BERT-SCORE | 0.8812 | **0.8891** | 0.8858 | 0.8871 | 0.8863 | 0.8867 | 0.8792 |
| **Topic Writing** | **0** | **0.2** | **0.5** | **1** | **1.5** | **2** | **5** |
| ROUGE-1 | 0.2916 | 0.3108 | **0.3139** | 0.3050 | 0.3035 | 0.3036 | 0.2820 |
| ROUGE-L | 0.1345 | 0.1365 | 0.1362 | 0.1359 | **0.1371** | 0.1349 | 0.1327 |
| BLEU | 3.8876 | 4.1522 | 4.0135 | 4.3646 | **4.4095** | 4.0991 | 2.8485 |
| METEOR | 0.2013 | 0.2363 | **0.2408** | 0.2253 | 0.2203 | 0.2308 | 0.1858 |
| BERT-SCORE | 0.7067 | 0.7181 | **0.7205** | 0.7150 | 0.7156 | 0.7142 | 0.7033 |

## Q  DIFFICULTY-AWARE EVALUATION FOR PERSONALIZED TASKS

To more comprehensively assess performance within the personalized generation, we constructed a difficulty-controlled evaluation experiment.

For each test sample, we first computed multiple automatic metrics of the base Instruct model (ROUGE, BLEU, METEOR, BERTScore). After normalization and aggregation, we ranked all samples and partitioned the test set into four difficulty levels.

As shown in Table 11, FlyThinker consistently surpasses SFT across all difficulty levels, including the hardest level (H0). This demonstrates that the dynamic reasoning mechanism remains highly effective even when the personalized generation task becomes more challenging.

Table 11: Difficulty-aware evaluation results for personalized generation. Test samples are grouped into four difficulty levels based on normalized automatic metric scores of the base Instruct model.

| | **H0 – Very Hard** | | **H1 – Hard** | | **H2 – Medium** | | **H3 – Easy** | |
|---|---|---|---|---|---|---|---|---|
| | SFT | FlyThinker | SFT | FlyThinker | SFT | FlyThinker | SFT | FlyThinker |
| ROUGE-1 | 0.2796 | **0.2923** | 0.3224 | **0.3390** | 0.3480 | **0.3679** | 0.3818 | **0.3965** |
| ROUGE-L | 0.1430 | **0.1441** | 0.1588 | **0.1612** | 0.1686 | **0.1714** | 0.1917 | **0.1932** |
| BLEU | 2.2154 | **2.3545** | 2.5867 | **2.8538** | 3.1460 | **3.5514** | 5.1719 | **5.5257** |
| METEOR | 0.2017 | **0.2275** | 0.2156 | **0.2443** | 0.2225 | **0.2494** | 0.2478 | **0.2649** |
| BERT-SCORE | 0.7175 | **0.7247** | 0.7793 | **0.7899** | 0.8080 | **0.8127** | 0.8396 | **0.8490** |

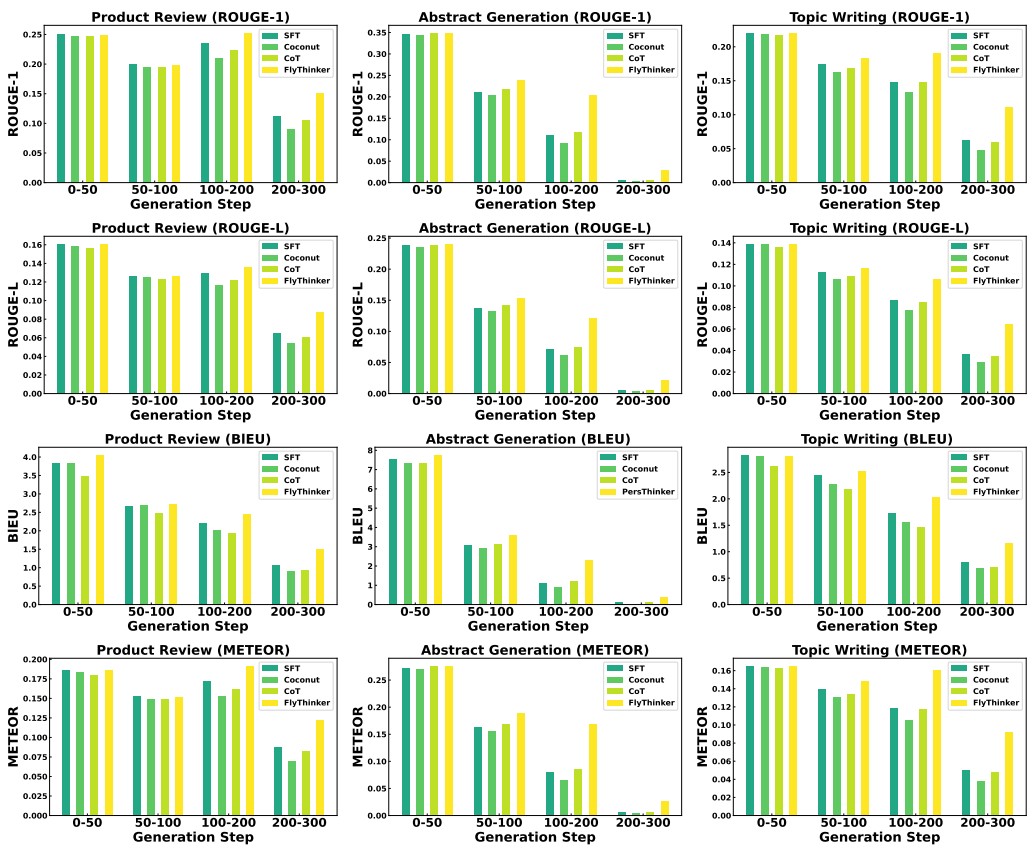

Figure 8: Position-sensitive evaluation across three datasets. Each generated sequence is divided into four segments by token position.

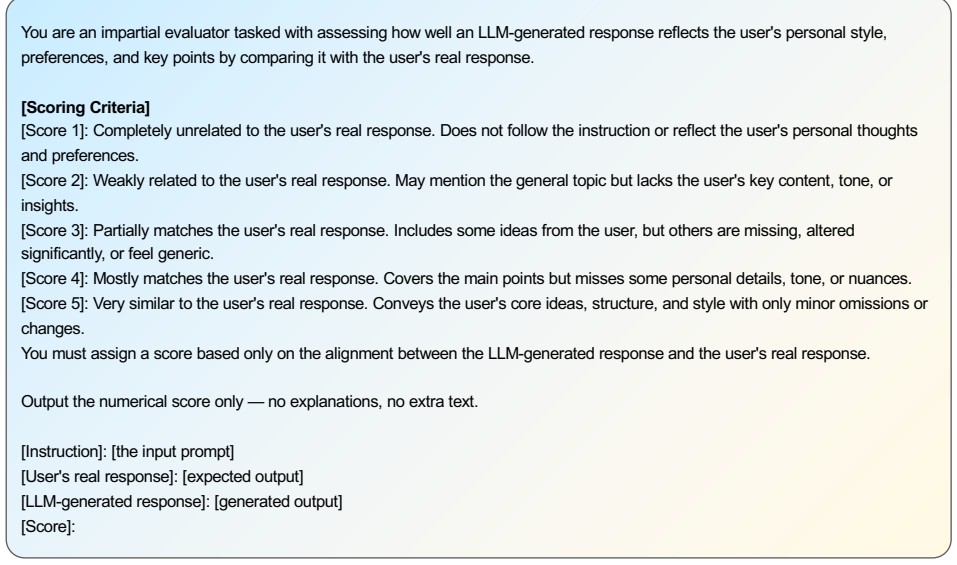

Figure 9: Illustration of the instruction template provided to GPT-4o-mini for conducting LLM-based personalized evaluation.

Table 12: Comprehensive comparison results with BERTScore as an additional metric and Average Relative Improvement (computed relative to Non-pers).

| Datasets | | Methods | Rouge-1 | Rouge-L | BLEU | Meteor | BERTScore | Impr. (↑) |
|---|---|---|---|---|---|---|---|---|
| **Product Review** | Tuning free | Non-pers | 0.2839 | 0.1317 | 1.5357 | 0.1783 | 0.7380 | - |
| | | RAG | 0.3176 | 0.1420 | 3.2990 | 0.2309 | 0.7482 | 33.08% |
| | | CoS | 0.2781 | 0.1347 | 2.9677 | 0.2189 | 0.7213 | 22.80% |
| | Tuning based | SFT | 0.3551 | 0.1536 | 3.9112 | 0.2309 | 0.7681 | 45.99% |
| | | LLM-TRSR | 0.3377 | 0.1450 | 2.3279 | 0.2122 | 0.7573 | 20.45% |
| | | NextQuill | 0.3311 | 0.1518 | 2.6599 | 0.2172 | 0.7580 | 25.92% |
| | | CoT | 0.3466 | 0.1508 | 3.3705 | 0.2220 | 0.7587 | 36.68% |
| | | Coconut | 0.3381 | 0.1496 | 3.3213 | 0.2093 | 0.7578 | 33.80% |
| | Ours | **FlyThinker** | **0.3663** | **0.1560** | **4.3620** | **0.2489** | **0.7755** | 55.24% |
| **Abstract Generation** | Tuning free | Non-pers | 0.3276 | 0.1717 | 4.5814 | 0.2476 | 0.8616 | - |
| | | RAG | 0.3168 | 0.1616 | 3.4047 | 0.2998 | 0.8818 | -2.28% |
| | | CoS | 0.2973 | 0.1576 | 3.2591 | **0.3023** | 0.8952 | -4.06% |
| | Tuning based | SFT | 0.3602 | 0.2062 | 5.8189 | 0.2392 | 0.8812 | 11.18% |
| | | LLM-TRSR | 0.3583 | 0.2056 | 5.5359 | 0.2371 | 0.8819 | 9.61% |
| | | NextQuill | 0.3501 | 0.2038 | 5.4388 | 0.2310 | 0.8737 | 7.79% |
| | | CoT | 0.3685 | 0.2071 | 5.8538 | 0.2469 | **0.9023** | 13.06% |
| | | Coconut | 0.3542 | 0.2041 | 5.2378 | 0.2298 | 0.8812 | 7.28% |
| | Ours | **FlyThinker** | **0.3716** | **0.2090** | **6.3383** | 0.2549 | 0.8858 | 15.85% |
| **Topic Writing** | Tuning free | Non-pers | 0.2389 | 0.1057 | 1.1190 | 0.1869 | 0.6986 | - |
| | | RAG | 0.2533 | 0.1099 | 1.4339 | 0.2253 | 0.6966 | 11.68% |
| | | CoS | 0.2307 | 0.1056 | 1.5426 | 0.2134 | 0.6891 | 9.42% |
| | Tuning based | SFT | 0.2916 | 0.1345 | 3.8876 | 0.2013 | 0.7067 | 61.11% |
| | | LLM-TRSR | 0.2839 | 0.1280 | 1.5552 | 0.1922 | 0.7040 | 16.50% |
| | | NextQuill | 0.2755 | 0.1269 | 2.1006 | 0.1817 | 0.6938 | 23.92% |
| | | CoT | 0.2846 | 0.1304 | 3.0020 | 0.1967 | 0.7138 | 43.63% |
| | | Coconut | 0.2786 | 0.1317 | 3.0736 | 0.1823 | 0.7012 | 42.76% |
| | Ours | **FlyThinker** | **0.3139** | **0.1362** | **4.0606** | **0.2409** | **0.7205** | 71.03% |

