# OpenReview forum: "Think-While-Generating: On-the-Fly Reasoning for Personalized Long-Form Generation"
_ICLR.cc/2026/Conference — ICLR 2026 Poster_

### Official Review · Reviewer_tgii · 2025-10-28

**Soundness:** 2
**Presentation:** 2
**Contribution:** 2
**Rating:** 4
**Confidence:** 4

**Summary:**

This paper introduces FlyThinker, an efficient “think-while-generating” framework. Unlike prior “reason-then-generate” approaches, FlyThinker employs a separate reasoning model that runs in parallel to dynamically guide the generation model. The framework achieves efficiency in both training and inference, and extensive experiments demonstrate its strong effectiveness and superior performance.

**Strengths:**

1. The concept of generating reasoning tokens and response tokens simultaneously is novel and intriguing to me.

2. The idea of decoupling reasoning tokens from previously generated reasoning outputs is particularly interesting, as it enables a one-pass training process and significantly improves overall efficiency.

**Weaknesses:**

1. The title of Figure 3 is somewhat misleading and should be revised to “Training Efficiency / Inference Efficiency.” Although the proposed method demonstrates shorter runtime, it relies on two separate models—the reasoning model and the generation model—which substantially increases memory consumption. The authors should therefore provide a comparison of the actual computational cost against other baselines to present a fair assessment.
2. Reasoning models typically show the greatest advantages on more challenging tasks, such as mathematical or scientific reasoning (e.g., AIME24, GPQA) and coding tasks. It would be more insightful if the authors evaluated the proposed method on such demanding benchmarks, as this would more convincingly highlight the true effectiveness of the reasoning model.

Typo: In line 94, the author redundantly includes an extra “First” after “Firstly.”

**Questions:**

As noted in the weaknesses above, please provide more details on the actual overall computational cost and evaluate the proposed method on a broader range of challenging tasks, such as mathematical or scientific reasoning and coding benchmarks, to better demonstrate its effectiveness.

---

> ### Author Response · Authors · 2025-11-22
> **Response to Reviewer tgii (Part 1)**
>
> Dear reviewer **tgii**,
>
> Thank you for your insightful review of our paper. Below we provide our detailed responses, and we hope they address your concerns.
>
>
> > **Q1:** The title of Figure 3 is somewhat misleading and should be revised to “Training Efficiency / Inference Efficiency.” Although the proposed method demonstrates shorter runtime, it relies on two separate models—the reasoning model and the generation model—which substantially increases memory consumption. The authors should therefore provide a comparison of the actual computational cost against other baselines to present a fair assessment.
>
> **Q1: Efficiency analysis.**
>
> **A1:** We have included a more comprehensive evaluation of memory, computational, and time-latency costs, summarized in the table below. The symbols are defined as:
>
> ● ($\theta_G$ / $\theta_R$): parameters of the generator and reasoner, respectively
>
> ● ($N_1$ / $N_2$): number of text tokens / reasoning tokens to generate
>
> ● ($C_G$ / $C_R$): cost of one forward pass for the generator / reasoner
>
> ● ($\ell_G$ / $\ell_R$): latency to generate one token by the generator / reasoner
>
> Specifically, regarding computational cost you focused: our method’s cost is approximately ($N_1 C_G + N_2 C_R$), while the baseline latent reasoning method incurs ($N_1 C_G + N_2 C_G$). The difference comes from ($N_2 C_R$) versus ($N_2 C_G$). Since the reasoner is typically much smaller than the generator (e.g., 0.5B vs. 7B), our method’s computational cost is significantly lower than latent reasoning.
>
> Compared to the SFT baseline (cost ($N_1 C_G$)), our additional cost is ($N_2 C_R$). As  ($\mathcal{C}_R \ll C_G$), this overhead is relatively small. For instance, using a 0.5B reasoner for a 7B generator with ($N_1 = N_2 = 1000$), the FLOPS are: our method — 15 TFLOPs, Cocount—28 TFLOPs, SFT — 14 TFLOPs. In this scenario, our method adds only 1 TFLOPs computation cost while delivering significant performance improvements, compared to SFT.
>
> Importantly, a key advantage of our method is reduced latency, as also shown in the table.
>
> | **Method**    | **Memory**                  | **Computation**                  | **Latency (Training stage)** | **Latency (Inference stage)** |
> |---------------|:-----------------------------:|:----------------------------------:|:-------------------------------:|:--------------------------------:|
> | SFT           | $\|\theta_{G}\|$              | $N_1 C_{G}$                      | $\ell_{G}$                   | $N_1\ell_{G}$                  |
> | CoT           | $\|\theta_{G}\|$              | $N_1 C_{G}+N_2 C_{G}$             | $\ell_{G}+N_2\ell_{G}$       | $N_1\ell_{G}+N_2\ell_{G}$      |
> | Coconut       | $\|\theta_{G}\|$              | $N_1 C_{G}+N_2 C_{G}$              | $\ell_{G}+N_2\ell_{G}$       | $N_1\ell_{G}+N_2\ell_{G}$      |
> | FlyThinker    | $\|\theta_{G}\| + \|\theta_{R}\|$ | $N_1 C_{G}+N_2 C_{R}$           | $\ell_G+\ell_{R}$            | $\approx N_1\ell_{G}$          |

---

> ### Author Response · Authors · 2025-11-22
> **Response to Reviewer tgii (Part 2)**
>
> > **Q2:** Reasoning models typically show the greatest advantages on more challenging tasks, such as mathematical or scientific reasoning (e.g., AIME24, GPQA) and coding tasks. It would be more insightful if the authors evaluated the proposed method on such demanding benchmarks, as this would more convincingly highlight the true effectiveness of the reasoning model.
>
> **Q2: More challenging tasks, like mathematical or scientific reasoning (e.g., AIME24, GPQA) and coding.**
>
> **A2:** The proposed FlyThinker framework is specifically designed for personalized long-form text generation, enabling the model’s reasoning to better align with the natural progression of user thinking during content creation.
>
> In response to your comments regarding model performance on challenging tasks, we first ranked our data into four difficulty levels based on baseline performance and then evaluated how our method performs on samples of varying difficulty. The results are presented in the table below. As shown, FlyThinker consistently outperforms SFT across all difficulty levels, including the hardest level (H0). This demonstrates that the dynamic reasoning mechanism remains highly effective even as the personalized generation task becomes more challenging.
>
> |            | H0 – Very Hard |                     | H1 – Hard    |                     | H2 – Medium  |                     | H3 – Easy     |                     |
> | ---------- | -------------- | ------------------- | ------------ | ------------------- | ------------ | ------------------- | ------------- | ------------------- |
> |            | **SFT (H0)**   | **FlyThinker (H0)** | **SFT (H1)** | **FlyThinker (H1)** | **SFT (H2)** | **FlyThinker (H2)** | **SFT  (H3)** | **FlyThinker (H3)** |
> | ROUGE-1    | 0.2796         | **0.2923**          | 0.3224       | **0.3390**          | 0.3480       | **0.3679**          | 0.3818        | **0.3965**          |
> | ROUGE-L    | 0.1430         | **0.1441**          | 0.1588       | **0.1612**          | 0.1686       | **0.1714**          | 0.1917        | **0.1932**          |
> | BLEU       | 2.2154         | **2.3545**          | 2.5867       | **2.8538**          | 3.1460       | **3.5514**          | 5.1719        | **5.5257**          |
> | METEOR     | 0.2017         | **0.2275**          | 0.2156       | **0.2443**          | 0.2225       | **0.2494**          | 0.2478        | **0.2649**          |
> | BERT-SCORE | 0.7175         | **0.7247**          | 0.7793       | **0.7899**          | 0.8080       | **0.8127**          | 0.8396        | **0.8490**          |
>
>
> Additionally, we considered experiments on math and code datasets. However, we observed that after standard SFT on one corresponding datasets, model performance degraded compared to the initial model. This likely occurs because the initial model has already undergone extensive training on code and math with large-scale data and advanced training strategies, so simplified fine-tuning can lead to a performance drop. After incorporating our method, performance improved significantly, but it still did not surpass the original model. This suggests that standard SFT alone is insufficient for these tasks; instead, our method needs to be integrated into the full training pipeline for code or math tasks to achieve optimal results.
>
> |                             | AIME24 | LiveCodeBench | GPQA   |
> | --------------------------- |:------:|:-------------:|:------:|
> | DeepSeek-R1-Distill-Qwen-7B | 0.5667 | 0.2767        | 0.5152 |
> | w/ SFT                      | 0.0334 | 0.0714        | 0.1969 |
> | w/ FlyThinker (R=1.5B)      | 0.1000 | 0.1785        | 0.3283 |
>
> > **Q3:** Typo: In line 94, the author redundantly includes an extra “First” after “Firstly.”
>
> **A3:** Thank you for your detailed review. We will reivse it.

---

> ### Author Response · Authors · 2025-11-24
> **ICLR Rebuttal: Your Feedback Is Kindly Anticipated**
>
> Dear Reviewer,
>
> **We hope that our responses satisfactorily resolve the issues you raised and would appreciate any further comments you may have at your convenience.**
>
> Thank you for your thoughtful review and consideration.
>
> Warm regards,
>
> Authors of submission 8572

---

> ### Author Response · Authors · 2025-11-26
> **ICLR Rebuttal: We Kindly Await Your Feedback**
>
> Dear Reviewer **tgii**,
>
> **We hope our responses have resolved your concerns and welcome any further comments.**
>
> Thank you for your review and consideration.
>
> Warm regards,
>
> Authors of ICLR submission 8572

---

> ### Author Response · Authors · 2025-11-27
> **Looking Forward to Your Feedback**
>
> Dear Reviewer tgii,
>
> Thank you for your insightful review. We have added further experiments and analyses addressing all your points. If you have any additional suggestions or questions, please feel free to let us know.
>
> We appreciate your time and consideration.
>
> Best regards, The Authors

---

> ### Author Response · Authors · 2025-11-27
> **Looking Forward to Your Feedback**
>
> Dear Reviewer tgii,
>
> We hope our responses have addressed your concerns and greatly look forward to receiving any further feedback.
>
> Warm regards,
>
> Authors

---

### Official Review · Reviewer_uQ75 · 2025-10-28

**Soundness:** 2
**Presentation:** 3
**Contribution:** 3
**Rating:** 4
**Confidence:** 4

**Summary:**

The authors propose FlyThinker, a method for combining LLM reasoning and generation in a "think-while-generating" paradigm. In their approach, there are reasoner and generator models, where the reasoner is given the input and an in-progress generation, then produces a latent reasoning token that is fed to the generator to guide its response. They argue that this approach allows for efficient LLM personalization with latent reasoning.

**Strengths:**

- The proposed approach seems useful for efficiently combining reasoning and generation. It provides a simple way to align to user preferences.
- The method is flexible and can be used with models of different sizes. It is particularly convenient that a small reasoning model can be combined with a larger generation model for greater efficiency.
- The writing and presentation of the paper are clear. The method section in particular is very easy to read and clearly lays out the method.

**Weaknesses:**

While the proposed approach is interesting, the evaluation experiments in their current form are not sufficiently comprehensive.
1.  The evaluation is based only on simple automated metrics (ROUGE, BLEU, METEOR, BERT-Score). To get a full understanding of how much FlyThinker improves personalization, it would be useful to have a user study or an automatic LLM evaluation of preferred personalized outputs.
2. The experiments are mostly limited to Qwen2.5-3B-Instruct. The small set of Qwen2.5-7B-Instruct experiments do not have any numbers on the axes, so it is difficult to tell how much FlyThinker improves performance over SFT. The experimental results would be strengthened by additional experiments with other models and filling out the Qwen2.5-7B-Instruct scores.
3. From my understanding, compared to SFT, with this approach it is necessary to keep up to 2x the number of parameters in memory. The authors state that their method is efficient because both the reasoner and generator can perform inference simultaneously, but do not discuss the added memory required at training and inference time. Some discussion of training and inference memory requirements (in addition to the runtime results already included) would be appreciated.

**Questions:**

- Figure 5 has no values in the axes, so it is hard to tell how significant the differences in scores actually are.
- Could the authors clarify and further fill out the results in Figure 6? The authors claim that "Moderate values (0.5-2) yield the best overall performance", but do not test any intermediate values between 0.5 and 2. Also, there seems to be some instability in this range, especially for abstract generation. There are no values on the y axis for this graph, so it is difficult to tell how much $\lambda$ affects performance.
- It would be valuable to see some examples of personalized outputs produced by FlyThinker, for example for different reasoning model sizes or $\lambda$ parameters. This would make it more clear how exactly these factors affect the outputs of the generator.

Minor typo: line 180: "tought" instead of "thought"

---

> ### Author Response · Authors · 2025-11-22
> **Response to Reviewer uQ75 (Part 1)**
>
> Dear reviewer **uQ75**,
>
> We sincerely appreciate the constructive feedback you provided. After careful consideration, we have addressed each of the points you raised and hope that our responses satisfactorily resolve them.
>
>
> > **Q1:**  The evaluation is based only on simple automated metrics (ROUGE, BLEU, METEOR, BERT-Score). To get a full understanding of how much FlyThinker improves personalization, it would be useful to have a user study or an automatic LLM evaluation of preferred personalized outputs.
>
> **Q1: Evaluation beyond simple automated metrics (ROUGE, BLEU, METEOR, BERT-Score)**
>
> **A1:** Following the suggestions, we added an LLM-based evaluation to provide a more comprehensive assessment of personalized generation quality. Specifically, we use GPT-4o as an evaluator and compare the outputs produced by FlyThinker with those from the strongest baseline, SFT, following a structured evaluation rubric (details will be included in the revised paper). As shown in the table below, FlyThinker consistently achieves higher scores across all datasets under the LLM-based evaluation. These findings indicate that FlyThinker’s personalization gains are reflected not only in traditional automatic metrics but also in the LLM judge’s assessments.
>
> |            | Product Review | Abstract Generation | Topic Writing |
> | ---------- | -------------- | ------------------- | ------------- |
> | SFT        | 0.3695         | 0.5718              | 0.2739        |
> | FlyThinker | **0.3816**     | **0.5955**          | **0.2854**    |
>
>
>
> > **Q2:**  The experiments are mostly limited to Qwen2.5-3B-Instruct. The small set of Qwen2.5-7B-Instruct experiments do not have any numbers on the axes, so it is difficult to tell how much FlyThinker improves performance over SFT. The experimental results would be strengthened by additional experiments with other models and filling out the Qwen2.5-7B-Instruct scores.
>
> **Q2: Filling out the Qwen2.5-7B-Instruct scores and adding experiments with other LLM backbones**
>
> **A2:** To address these concerns, we have: 1) completed the Qwen2.5-7B-Instruct results, and 2) added experiments using the Gemma-7B-it model. The results (compared with the best-performing baselines) are summarized in the tables below. From the results, we can clearly observe that:
>
> 1) On the Qwen2.5-7B-Instruct model, FlyThinker outperforms the representative baseline, SFT, on all datasets.
>
> 2) On the Gemma-7B-it model, FlyThinker also shows consistent performance gains. Together with the results on Qwen2.5-3B and 7B, these findings indicate that our approach demonstrates strong generalization across different model architectures.
>
> ### **Qwen2.5-7B-Instruct**
>
> | **Product Review** | **ROUGE-1** | **ROUGE-L** | **BLEU** | **METEOR** | **BERT-SCORE** |
> | :--- | :---: | :---: | :---: | :---: | :---: |
> | SFT | 0.3512 | 0.1560 | 3.8953 | 0.2224 | 0.7674 |
> | FlyThinker (R=1.5B) | **0.3738** | **0.1578** | **5.1072** | **0.2715** | **0.7846** |
> | **Abstract Generation** | **ROUGE-1** | **ROUGE-L** | **BLEU** | **METEOR** | **BERT-SCORE** |
> | SFT | 0.3645 | **0.2122** | 5.9661 | 0.2398 | 0.8854 |
> | FlyThinker (R=1.5B) | **0.3708** | 0.1971 | **6.2732** | **0.3010** | **0.8910** |
> | **Topic Writing** | **ROUGE-1** | **ROUGE-L** | **BLEU** | **METEOR** | **BERT-SCORE** |
> | SFT | 0.2849 | **0.1351** | 3.4850 | 0.1883 | 0.7071 |
> | FlyThinker (R=1.5B) | **0.3076** | 0.1325 | **3.6572** | **0.2564** | **0.7249** |
>
> ### **Gemma-7B-it**
> | **Product Review** | **ROUGE-1** | **ROUGE-L** | **BLEU** | **METEOR** | **BERT-SCORE** |
> | :--- | :---: | :---: | :---: | :---: | :---: |
> | SFT | 0.3285 | 0.1544 | **3.9774** | **0.2166** | **0.7559** |
> | FlyThinker (R=2B) | **0.3355** | **0.1579** | 3.5646 | 0.2160 | 0.7553 |
> | **Abstract Generation** | **ROUGE-1** | **ROUGE-L** | **BLEU** | **METEOR** | **BERT-SCORE** |
> | SFT | 0.3501 | 0.2057 | 6.1330 | 0.2404 | 0.8750 |
> | FlyThinker (R=2B) | **0.3624** | **0.2062** | **6.8307** | **0.2638** | **0.8800** |
> | **Topic Writing** | **ROUGE-1** | **ROUGE-L** | **BLEU** | **METEOR** | **BERT-SCORE** |
> | SFT | 0.2664 | 0.1323 | **3.9175** | 0.1938 | 0.6960 |
> | FlyThinker (R=2B) | **0.2829** | **0.1385** | 3.6046 | **0.2035** | **0.6989** |

---

> ### Author Response · Authors · 2025-11-22
> **Response to Reviewer uQ75 (Part 2)**
>
> > **Q3:**  From my understanding, compared to SFT, with this approach it is necessary to keep up to 2x the number of parameters in memory. The authors state that their method is efficient because both the reasoner and generator can perform inference simultaneously, but do not discuss the added memory required at training and inference time. Some discussion of training and inference memory requirements (in addition to the runtime results already included) would be appreciated.
>
>
> **Q3: Memory Cost**
>
> **A3:** We have included a more comprehensive evaluation of memory, computational, and time-latency costs, summarized in the table below. Regarding memory, the cost of is related to ($|\theta_G| + |\theta_R|$). Compared to the baseline, the additional cost comes from ($|\theta_R|$), which depends on the reasoner’s size. In experiments, a 0.5B reasoner is sufficient for a 7B generator to obtain good results (see Reviewer vYSZ Q2). Thus, the memory overhead can be minimal in real applications. Please note that the core advantage of our method is to reduce both inference and training latency, a core challenge for latent reasoning methods.
>
> ● ($\theta_G$ / $\theta_R$): parameters of the generator and reasoner, respectively
>
> ● ($N_1$ / $N_2$): number of text tokens / reasoning tokens to generate
>
> ● ($C_G$ / $C_R$): cost of one forward pass for the generator / reasoner
>
> ● ($\ell_G$ / $\ell_R$): latency to generate one token by the generator / reasoner
>
>
> | **Method**    | **Memory**                  | **Computation**                  | **Latency (Training)** | **Latency (Inference)** |
> |---------------|:-----------------------------:|:----------------------------------:|:-------------------------------:|:--------------------------------:|
> | SFT           | $\|\theta_{G}\|$              | $N_1 C_{G}$                      | $\ell_{G}$                   | $N_1\ell_{G}$                  |
> | CoT           | $\|\theta_{G}\|$              | $N_1 C_{G}+N_2 C_{G}$             | $\ell_{G}+N_2\ell_{G}$       | $N_1\ell_{G}+N_2\ell_{G}$      |
> | Coconut       | $\|\theta_{G}\|$              | $N_1 C_{G}+N_2 C_{G}$              | $\ell_{G}+N_2\ell_{G}$       | $N_1\ell_{G}+N_2\ell_{G}$      |
> | FlyThinker    | $\|\theta_{G}\| + \|\theta_{R}\|$ | $N_1 C_{G}+N_2 C_{R}$           | $\ell_G+\ell_{R}$            | $\approx N_1\ell_{G}$          |
>
> > **Q4:** Figure 5 has no values in the axes, so it is hard to tell how significant the differences in scores actually are.
>
> **A4:** Sorry for forgetting including the detailed values, will include them in our revised paper. Here are the detailed resutls:
>
> | Product Review          | ROUGE-1     | ROUGE-L     | BLEU     | METEOR     |
> | ----------------------- | ----------- | ----------- | -------- | ---------- |
> | SFT                     | 0.3551      | 0.1536      | 3.9112   | 0.2309     |
> | FlyThinker (R=3B)       | 0.3663      | 0.1560      | 4.3620   | 0.2489     |
> | FlyThinker (R=1.5B)     | 0.3676      | 0.1554      | 4.2123   | 0.2501     |
> | FlyThinker (R=0.5B)     | 0.3556      | 0.1531      | 3.8072   | 0.2366     |
> | **Abstract Generation** | **ROUGE-1** | **ROUGE-L** | **BLEU** | **METEOR** | **BERT-SCORE** |
> | SFT                     | 0.3602      | 0.2062      | 5.8189   | 0.2392     |
> | FlyThinker (R=3B)       | 0.3716      | 0.2090      | 6.3383   | 0.2549     |
> | FlyThinker (R=1.5B)     | 0.3726      | 0.1989      | 6.2690   | 0.2910     |
> | FlyThinker (R=0.5B)     | 0.3603      | 0.1915      | 5.5334   | 0.2985     |
> | **Topic Writing**       | **ROUGE-1** | **ROUGE-L** | **BLEU** | **METEOR** | **BERT-SCORE** |
> | SFT                     | 0.2916      | 0.1345      | 3.8876   | 0.2013     |
> | FlyThinker (R=3B)       | 0.3139      | 0.1362      | 4.0606   | 0.2409     |
> | FlyThinker (R=1.5B)     | 0.3115      | 0.1348      | 3.8004   | 0.2450     |
> | FlyThinker (R=0.5B)     | 0.2960      | 0.1297      | 3.2908   | 0.2450     |

---

> ### Author Response · Authors · 2025-11-22
> **Response to Reviewer uQ75 (Part 3)**
>
> > **Q5:** Could the authors clarify and further fill out the results in Figure 6? The authors claim that "Moderate values (0.5-2) yield the best overall performance", but do not test any intermediate values between 0.5 and 2. Also, there seems to be some instability in this range, especially for abstract generation. There are no values on the y axis for this graph, so it is difficult to tell how much \lambda affects performance.
>
> **Q5: Hyper-parameter analysis. Do not test any intermediate values between 0.5 and 2**
>
> **A5:**  To address the concerns, we have added two additional λ values, 1.0 and 1.5, expanding the tested set to {0, 0.2, 0.5, 1.0, 1.5, 2.0}. The results are summarized in the following table, from which we make the following observations: 1) Fine-grained λ tuning provides additional performance gains. 2) When λ is around 1.0—neither too small nor too large—the model consistently outperforms the SFT baseline. 3) When λ varies within [0.5, 2.0], individual metrics fluctuate slightly, but all results remain above the SFT baseline, indicating that our method’s effectiveness is relatively stable within this range.
>
> | **Product Review**      | **λ = 0** | **λ = 0.2** | **λ = 0.5** | **λ = 1**  | **λ = 1.5** | **λ = 2**  | **λ = 5** |
> | ----------------------- | --------- | ----------- | ----------- | ---------- | ----------- | ---------- | --------- |
> | ROUGE-1                 | 0.3551    | 0.3641      | **0.3664**  | 0.3551     | 0.3569      | 0.3626     | 0.3271    |
> | ROUGE-L                 | 0.1536    | 0.1554      | **0.1560**  | 0.1546     | 0.1543      | 0.1550     | 0.1479    |
> | BLEU                    | 3.9112    | 4.1088      | 4.3223      | 4.0979     | 4.0685      | **4.6599** | 2.7995    |
> | METEOR                  | 0.2309    | 0.2419      | 0.2489      | 0.2334     | 0.2353      | **0.2529** | 0.1976    |
> | BERT-SCORE              | 0.7681    | 0.7723      | **0.7755**  | 0.7687     | 0.7688      | 0.7740     | 0.7541    |
> | **Abstract Generation** | **λ = 0** | **λ = 0.2** | **λ = 0.5** | **λ = 1**  | **λ = 1.5** | **λ = 2**  | **λ = 5** |
> | ROUGE-1                 | 0.3602    | **0.3750**  | 0.3716      | 0.3748     | 0.3729      | 0.3734     | 0.3540    |
> | ROUGE-L                 | 0.2062    | 0.2016      | **0.2090**  | 0.2068     | 0.2081      | 0.2077     | 0.2022    |
> | BLEU                    | 5.8189    | 6.5881      | 6.3383      | **7.0687** | 6.7091      | 6.8036     | 5.4578    |
> | METEOR                  | 0.2392    | **0.2891**  | 0.2549      | 0.2718     | 0.2633      | 0.2657     | 0.2345    |
> | BERT-SCORE              | 0.8812    | **0.8891**  | 0.8858      | 0.8871     | 0.8863      | 0.8867     | 0.8792    |
> | **Topic Writing**       | **λ = 0** | **λ = 0.2** | **λ = 0.5** | **λ = 1**  | **λ = 1.5** | **λ = 2**  | **λ = 5** |
> | ROUGE-1                 | 0.2916    | 0.3108      | **0.3139**  | 0.3050     | 0.3035      | 0.3036     | 0.2820    |
> | ROUGE-L                 | 0.1345    | 0.1365      | 0.1362      | 0.1359     | **0.1371**  | 0.1349     | 0.1327    |
> | BLEU                    | 3.8876    | 4.1522      | 4.0135      | 4.3646     | **4.4095**  | 4.0991     | 2.8485    |
> | METEOR                  | 0.2013    | 0.2363      | **0.2408**  | 0.2253     | 0.2203      | 0.2308     | 0.1858    |
> | BERT-SCORE              | 0.7067    | 0.7181      | **0.7205**  | 0.7150     | 0.7156      | 0.7142     | 0.7033    |

---

> ### Author Response · Authors · 2025-11-22
> **Response to Reviewer uQ75 (Part 4)**
>
> > **Q6:** It would be valuable to see some examples of personalized outputs produced by FlyThinker, for example for different reasoning model sizes or $\lambda$ parameters. This would make it more clear how exactly these factors affect the outputs of the generator.
>
> **Q6: Case study for results with diferent reasoning model sizes**
>
> **A6:** Thank you for the suggestion. We have added a case study. Below is a comparison of the results using the 3B and 0.5B Reasoners:
>
> **Question:** Generate the review...for a product with description "Based on... breaks the world one-hour record on ...". The ... is "... more about a battle with depression...".'
>
> **User's Real Writing:**  ''...... using a home-made bicycle (even using parts from a washing machine) (1) breaks the world\'s endurance record in a time trial that had cyclists everywhere in awe.(2) ......, takes Obree to the point of suicide, and a long wrestle with manic depression.......It was a painful lack of self-confidence instilled by years of bullying and by precious little help from his own father, a policeman. (3) As with many trues stories of depression,(4) ......If you\'re looking for an exciting cycle race movie, no, this is not the one for you, but if you want a movie that takes you into the sometimes dark world of the human soul, be prepared for one tough ride.(5) ......As this inspirational story shows: a world record didn\'t satisfy Obree\'s demons. ......(6)
>
> **Analysis:** The user's real writing includes the following elements: factual information, such as (1) bicycle production and (2) awards; emotional tone, such as (3), which is predominantly serious and somber, in line with the story's theme; writing techniques, including analogy (4), contrast (5), and (6); and core perspectives, such as (4), which emphasizes that the central focus of the narrative is the protagonist's internal struggle rather than a competition, and (5), which underscores that the protagonist's inner world holds more significance than external honors.
>
> **FlyThinker (R=3B):** ...... broke the world one-hour cycling record (1) ......It's not an Olympic film: it focuses on the battle between Graeme and depression (2)......Graeme is a complex and fascinating human being who has a remarkable ability to focus and achieve, but he has a fragile and troubled mind.(3) ...... There are parallels to someone like Lance Armstrong (who also battled depression as a result of an accident.)(4) This movie is not about racing or cycling ...... it's about the pursuit of excellence - but it's more about the pursuit of mental sanity.(5)
>
> **Analysis:** The generated content contains many characteristics consistent with the user's real writing: it incorporates key factual details (1), and maintains an emotional tone that is serious, somber, and centered on depression (3). Additionally, the writing techniques are consistent, utilizing contrast (2), (3), (5), and analogy (3). Furthermore, the core perspectives are aligned, such as (2), which highlights that the story revolves around the protagonist's internal conflict rather than a competition, and (5), which emphasizes the primacy of the protagonist's inner world over external honors.
>
> **FlyThinker (R=0.5B):** ...... this story of one man\'s extraordinary battle with depression.(1)...... a frame of welded together steel pipes, some washers, two broken front wheels, and one broken pedal.(2) ...... I had seen some photos of Graeme cycling around Glasgow, and I had read the article about his cycling career in the Sunday Herald back in the late 1980s.(3) ...... One small gripe: the ending of the film, which is a replay of the race, feels too abrupt. ...... '(4)
>
> **Analysis:** The generated content contains some features that align with the user's writing, such as mentioning similar factual information (2) and introducing the theme of the story, as in (1), which emphasizes the protagonist's internal struggle rather than a competition. However, the content also includes a significant amount of irrelevant information (3), (4), and does not maintain the serious, somber emotional tone. Instead, it adopts a lighter emotional approach (3) and fails to employ the user's typical writing techniques.
>
> Our results show that larger Reasoner models better capture the user’s tone, emotional depth, and narrative structure better. The full case study will be provided in the Appendix.
>
> > **Q7:** Minor typo: line 180: "tought" instead of "thought"
>
> **A7:** Thanks. We will reivse it.

---

> ### Author Response · Authors · 2025-11-24
> **ICLR Rebuttal: Your Feedback Would Be Greatly Appreciated**
>
> Dear Reviewer,
>
> We have updated our manuscript. In particular, the full version of the case study related to **Question 3** is now included in Appendix Q.
>
> **We hope that our responses satisfactorily resolve the issues you raised and kindly welcome any further feedback at your earliest convenience.**
>
> Thank you very much for your time and careful evaluation.
>
> Sincerely,
>
> Authors of submission 8572

---

> ### Author Response · Authors · 2025-11-26
> **ICLR Rebuttal: Looking Forward to Hearing from You**
>
> Dear Reviewer **uQ75**,
>
> We fully understand the significant time constraints and responsibilities faced by you during this critical period.
>
> **As the decision phase approaches, we would greatly appreciate any post-rebuttal comments you might share to help ensure a fair and thorough final evaluation of our submission.**
>
> We hope our responses have addressed your concerns and welcome any further feedback.
>
>
> Warm regards,
>
> Authors of ICLR submission 8572

---

> ### Author Response · Authors · 2025-11-27
> **Looking Forward to Your Feedback**
>
> Dear Reviewer uQ75,
>
> Thank you for your insightful review. We have added further experiments and analyses addressing all your points. If you have any additional suggestions or questions, please feel free to let us know.
>
> We appreciate your time and consideration.
>
> Best regards, The Authors

---

> ### Author Response · Authors · 2025-11-27
> **Looking Forward to Your Feedback**
>
> Dear Reviewer uQ75,
>
> We hope our responses have addressed your concerns and greatly look forward to receiving any further feedback.
>
> Warm regards,
>
> Authors

---

### Official Review · Reviewer_vYSZ · 2025-10-29

**Soundness:** 3
**Presentation:** 3
**Contribution:** 2
**Rating:** 6
**Confidence:** 3

**Summary:**

This paper proposes FlyThinker, a “think-while-generating” framework for personalized long-form text generation. FlyThinker employs a Reasoner that generates latent token-level reasoning signals and a Generator that dynamically integrates these reasoning signals into its token-level predictions, enabling parallel training and inference. Experiments on three tasks from the LONGLAMP benchmark, including Product Review, Abstract Generation, and Topic Writing, demonstrate that FlyThinker achieves improvements in both personalization quality and generation efficiency over several baselines.

**Strengths:**

1. The “think-while-generating” paradigm is well-motivated, and the overall methodology is simple and intuitive.
2. The evaluation is comprehensive with multiple metrics, showing the effectiveness of the proposed method over baselines.
3. FlyThinker achieves training and inference efficiency comparable to SFT, which is a major advantage relative to existing reasoning-augmented methods that typically incur higher latency.

**Weaknesses:**

1. It is not clear whether the reported personalization results are based on the user-based split (testing on unseen users) or the temporal split (testing on later instances of seen users). Since these settings test different personalization abilities (cross-user vs. within-user), clarification or stratified results would make the findings more interpretable.

2. While the paper ablates on Reasoner size, it is not clear how the Reasoner scales relative to the Generator, e.g., what would be the smallest Reasoner that still remains effective for different Generator sizes - would it be roughly 30%, or 50% of the Generator size? Insights into this would provide very helpful practical guidance for applying FlyThinker in real-world settings.

3. Appendix H shows that adding reasoning tokens to both input and output positions yields the best performance. It is not clear how sensitive FlyThinker is to user history length, i.e., when each user has a lot of historical records with long-form generations, making the context very long. A discussion on this would strengthen the paper’s empirical insights.

4. Are the learned latent reasoning tokens interpretable, or could they be used for downstream applications such as user clustering or user preference visualization? Some discussions on this would offer insights into the interpretability of the latent reasoning tokens.

**Questions:**

Please refer to Weaknesses.

---

> ### Author Response · Authors · 2025-11-22
> **Response to Reviewer vYSZ (Part 1)**
>
> Dear reviewer **vYSZ**,
>
> Thank you for your insightful review of our paper. We have given careful thought to each of your comments and have provided our responses below to address your concerns.
>
> > **Q1:**  It is not clear whether the reported personalization results are based on the user-based split (testing on unseen users) or the temporal split (testing on later instances of seen users). Since these settings test different personalization abilities (cross-user vs. within-user), clarification or stratified results would make the findings more interpretable.
>
> **Q1: Clarification on the dataset split method — user-based or temporal split?**
>
> **A1:**  In our experiments, we adopt a user-based split.
>
> >**Q2:**  While the paper ablates on Reasoner size, it is not clear how the Reasoner scales relative to the Generator, e.g., what would be the smallest Reasoner that still remains effective for different Generator sizes - would it be roughly 30%, or 50% of the Generator size? Insights into this would provide very helpful practical guidance for applying FlyThinker in real-world settings.
>
> **Q2: What is the smallest Reasoner that remains effective for different Generator sizes?**
>
> **A2:** To answer this question, we conducted a systematic analysis using 3B and 7B Generators, and tested Reasoners with sizes of 0.5B and 1.5B. The results are presented in the table below. As shown, with a 7B Generator, a 0.5B Reasoner can already deliver performance improvements, while with a 3B Generator, the Reasoner needs to reach 1.5B to achieve meaningful and consistent gains across metrics. These results suggest that a generally small Reasoner (around 1.5B) can be effective. However, there is no clear proportional size requirement between the Reasoner and the Generator — a stronger Generator places fewer demands on the Reasoner. Naturally, a larger Reasoner typically delivers better performance.
>
>
> | | **G = 3B** | | | | | **G = 7B** | | | | |
> | :--- | :---: | :---: | :---: | :---: | :---: | :---: | :---: | :---: | :---: | :---: |
> |  | **ROUGE-1** | **ROUGE-L** | **BLEU** | **METEOR** | **BERT-SCORE** | **ROUGE-1** | **ROUGE-L** | **BLEU** | **METEOR** | **BERT-SCORE** |
> | SFT | 0.3551 | **0.1536** | **3.9112** | 0.2309 | 0.7681 | 0.3512 | 0.1560 | 3.8953 | 0.2224 | 0.7674 |
> | FlyThinker (R=1.5B) | **0.3676** | **0.1554** | **4.2123** | **0.2501** | **0.7729** | **0.3738** | **0.1578** | **5.1072** | **0.2715** | **0.7846** |
> | FlyThinker (R=0.5B) | **0.3556** | 0.1531 | 3.8072 | **0.2366** | **0.7692** | **0.3712** | **0.1566** | **4.8035** | **0.2737** | **0.7813** |

---

> ### Author Response · Authors · 2025-11-22
> **Response to Reviewer vYSZ (Part 2)**
>
> > **Q3:**  Appendix H shows that adding reasoning tokens to both input and output positions yields the best performance. It is not clear how sensitive FlyThinker is to user history length, i.e., when each user has a lot of historical records with long-form generations, making the context very long. A discussion on this would strengthen the paper’s empirical insights.
>
> **Q3: The sensitivity analysis to user history length.**
>
> **A3:** To address these concerns, we analyzed FlyThinker’s performance under different history lengths. Specifically, we tested three settings: two history examples (～1,750 tokens), three history examples (～2,250 tokens), and four history examples (～2,750 tokens), comparing our method with the SFT method. The results, summarized in the table below, show that performance increases as the context length grows across most metrics, while FlyThinker consistently outperforms SFT in all tested settings. These findings suggest that our method is relatively insensitive to user history length (if not exceed the context limits of LLMs).
>
> | | **#his=2, #his_token≈1750** | | **#his=3, #his_token≈2250** | | **#his=4, #his_token≈2750** | |
> | :--- | :---: | :---: | :---: | :---: | :---: | :---: |
> | **Product Review** | **SFT** | **FlyThinker** | **SFT** | **FlyThinker** | **SFT** | **FlyThinker** |
> | ROUGE-1 | 0.3551 | 0.3663 | 0.3619 | 0.3705 | 0.3635 | 0.3719 |
> | ROUGE-L | 0.1536 | 0.1560 | 0.1568 | 0.1581 | 0.1585 | 0.1619 |
> | BLEU | 3.9112 | 4.3620 | 4.4810 | 4.5020 | 5.1703 | 5.0671 |
> | METEOR | 0.2309 | 0.2489 | 0.2394 | 0.2490 | 0.2437 | 0.2511 |
> | BERT-SCORE | 0.7681 | 0.7755 | 0.7711 | 0.7762 | 0.7708 | 0.7772 |
> | **Abstract Generation** | **SFT** | **FlyThinker** | **SFT** | **FlyThinker** | **SFT** | **FlyThinker** |
> | ROUGE-1 | 0.3602 | 0.3716 | 0.3648 | 0.3786 | 0.3651 | 0.3795 |
> | ROUGE-L | 0.2062 | 0.2090 | 0.2095 | 0.2035 | 0.2102 | 0.2047 |
> | BLEU | 5.8189 | 6.3383 | 5.9950 | 6.8119 | 6.0234 | 7.0848 |
> | METEOR | 0.2392 | 0.2549 | 0.2422 | 0.2903 | 0.2432 | 0.2882 |
> | BERT-SCORE | 0.8812 | 0.8858 | 0.8845 | 0.8910 | 0.8852 | 0.8910 |
> | **Topic Writing** | **SFT** | **FlyThinker** | **SFT** | **FlyThinker** | **SFT** | **FlyThinker** |
> | ROUGE-1 | 0.2916 | 0.3139 | 0.2963 | 0.3153 | 0.2980 | 0.3185 |
> | ROUGE-L | 0.1345 | 0.1362 | 0.1360 | 0.1376 | 0.1359 | 0.1401 |
> | BLEU | 3.8876 | 4.0606 | 4.8021 | 4.4026 | 4.7817 | 4.8841 |
> | METEOR | 0.2013 | 0.2409 | 0.2084 | 0.2384 | 0.2126 | 0.2395 |
> | BERT-SCORE | 0.7067 | 0.7205 | 0.7092 | 0.7197 | 0.7091 | 0.7204 |
>
> > **Q4:**  Are the learned latent reasoning tokens interpretable, or could they be used for downstream applications such as user clustering or user preference visualization? Some discussions on this would offer insights into the interpretability of the latent reasoning tokens.
>
> **Q4: Interpretablity of the latent reasonign or the potential applicaitons.**
>
> **A4:** Interpretablity is common challenge for latent reasoning. Draw inspiration from your comments, we have conducted an visualization study. Under a fixed task, we selected five users and extracted one sample from each of them. We compared the Reasoner latent reasoning tokens for these samples and visualized them using t-SNE. We then used linear regression to estimate the direction from early to later reasoning tokens, representing each user’s reasoning trajectory.  The corresponding figure for this experiment illustrates that different users show different trajectory directions (This figure will be included in the revised version of the paper).
>
> To examine whether these trajectories relate to the preference, we used GPT-5.1 to compare the samples. We selected the two samples whose users had the most similar trajectory directions (sample0 and sample4) and the one farthest away (sample2). GPT-5.1 judged sample4 to be most similar to sample0 in paragraph structure, narrative style, and analytical approach. Sample2 was the least similar. These results show that different users induce distinct reasoning trajectories, and that closer trajectories correspond to more similar writing styles and semantic structures, etc. This suggests that latent reasoning tokens may contain interpretable semantic information useful for user clustering and preference visualization  (The results of this case study will be included in the revised version of the paper).

---

> ### Author Response · Authors · 2025-11-24
> **ICLR-Rebuttal: Looking Forward to Your Feedback**
>
> Dear Reviewer,
>
> We have updated our manuscript. In particular, the **visualization and interpretability** case study related to **Weakness 4** can be found in Appendix H.
>
> **We sincerely hope that our responses adequately address your concerns and would greatly appreciate any feedback at your convenience.**
>
> Thank you very much for your time and consideration.
>
> Sincerely,
>
> Authors of submission 8572

---

> ### Author Response · Authors · 2025-11-26
> **ICLR Rebuttal: Your Feedback Would Be Greatly Appreciated**
>
> Dear Reviewer **vYSZ**,
>
> We understand the significant time pressures you are facing at this critical stage.
>
> We hope our responses have addressed your concerns and greatly look forward to receiving any further feedback.
>
> Warm regards,
>
> Authors of ICLR submission 8572

---

> ### Author Response · Authors · 2025-11-27
> **Looking Forward to Your Feedback**
>
> Dear Reviewer vYSZ,
>
> Thank you for your insightful review. We have added further experiments and analyses addressing all your points. If you have any additional suggestions or questions, please feel free to let us know.
>
> We appreciate your time and consideration.
>
> Best regards, The Authors

---

> ### Author Response · Authors · 2025-11-27
> **Looking Forward to Your Feedback**
>
> Dear Reviewer vYSZ,
>
> We hope our responses have addressed your concerns and greatly look forward to receiving any further feedback.
>
> Warm regards,
>
> Authors

---

### Author Response · Authors · 2025-11-27
**Response Summary**

**Dear ACs and Reviewers,**

We are sincerely grateful for the time and insightful feedback provided throughout this process. We are encouraged by the reviewers’ recognition of our work’s:

1. **Strong Motivation and Novelty**: the paradigm is well-motivated (@vYSZ), novel and intriguing (@tgii), and particularly interesting (@tgii).
2. **Methodological Strength**: the method is simple and intuitive (@vYSZ) as well as flexible (@uQ75).
3. **Impressive Effectiveness and Efficiency**: the evaluation is comprehensive and a major advantage (@vYSZ), demonstrates greater efficiency (@uQ75), and significantly improves performance (@tgii).
4. **Clear Presentation**: the paper is clear and very easy to read (@uQ75).

---

---

**Summary of Major Clarifications and Revisions**

1. **Computation and Memory Cost Analysis**

 We now include a detailed comparison of memory usage, computational cost, and latency relative to other reasoning methods. Although FlyThinker incorporates two models, it introduces only minimal additional overhead (e.g. a 0.5B Reasoner) while substantially reducing latency and preserving high efficiency (@uQ75, @tgii).

2. **Strengthened Experimental Results**

 We expanded several analyses to better address reviewer feedback, including:

 ● LLM-based evaluation (@uQ75)

 ● Generalization across different LLM backbones (@uQ75)

 ● Hyperparameter analysis (@uQ75)

3. **New Empirical Insights and Case Studies**

 We incorporated or discussed the reviewer-suggested directions:

 ● Reasoner–Generator scaling (@vYSZ)

 ● Interpretability of latent reasoning tokens (@vYSZ)

 ● Sensitivity to user history (@vYSZ)

 ● A case study on Reasoner size (@uQ75)

4. **Future Exploration: Hard-Task Evaluation**

 We further analyzed FlyThinker’s performance across four difficulty levels. In addition, preliminary results on AIME24, GPQA, and LiveCodeBench indicate promising potential for extending FlyThinker to more challenging mathematical and coding-reasoning tasks, outlining a clear direction for future work (@tgii).

---

---

We provide point-by-point responses to each reviewer below. All corresponding modifications have been incorporated into the revised manuscript and highlighted in blue.

We sincerely hope that these responses and additional experimental results address all reviewer concerns and further strengthen the paper’s contributions.

---

### Author Response · Authors · 2025-12-03
**Summary of Reviews and Discussion**

**Dear Area Chair:**

We sincerely appreciate your meta-review of our paper, especially given the substantial additional workload caused by the OpenReview leakage. For your convenience, we provide a concise summary of the reviews and the discussion process.


We are highly encouraged by the reviewers’ recognition of our work’s strengths:

1. **Strong Motivation and Novelty**: The proposed paradigm is well-motivated (@vYSZ), novel and intriguing (@tgii), and particularly interesting (@tgii).

---

2. **Methodological Strength**: The method is described as simple and intuitive (@vYSZ) as well as flexible (@uQ75).

---

3. **Impressive Effectiveness and Efficiency**: The evaluation is comprehensive and a major advantage (@vYSZ), demonstrates greater efficiency (@uQ75), and achieves significantly improved performance (@tgii).

---

4. **Clear Presentation**: The paper is clear and very easy to read (@uQ75).

---

**Summary of Key Points and Revisions Per Reviewer**

**We were pleasantly surprised to observe a consistent pattern in the "weaknesses" highlighted by all three reviewers. Their comments were uniformly framed as suggestions, specifically: "The paper would be further improved by incorporating/supplementing…" This clearly indicates the high expectations the reviewers have for our work.**

---
---

**Reviewer 1: (@vYSZ)**

Reviewer 1 provided positive scores. Their questions included:

**1.Clarification on the dataset split method.**

This clarification has already been added to the revised paper.

**2.Suggested additional experiments:** Reasoner–Generator scaling, sensitivity to user history, interpretability of latent reasoning tokens.

These experiments and analyses were already included in our revision, showing that 1) a stronger Generator places fewer demands on the Reasoner (e.g., with a 7B Generator, a 0.5B Reasoner can already deliver performance improvements), 2) the performance increases as the history length grows, while our method consistently outperforms baseline in all tested settings, and 3)  latent reasoning tokens contain interpretable information useful for user clustering.

---
---

**Reviewer 2 (@uQ75)**

This reviewer’s main concerns were:

**1.Lack of clarity in certain figures.**

We have included detailed numerical results in the appendix to augment the corresponding figures in the main text.

**2.Recommendation to add more experiments:** memory cost, LLM-based evaluation, generalization across different LLM backbones, and case studies.

We have included all of these experiments in the revised paper and rebuttal, showing that 1) the memory cost remains low when the reasoner is small (e.g., a 0.5B reasoner combined with a 7B generator works effectively), 2) our method also outperforms baselines under LLM-based evaluation, and 3) it generalizes well across different LLM backbones.

---
---

**Reviewer 3 (@tgii)**

This reviewer found our method **highly appealing and expressed interest** in seeing evaluations on more challenging tasks, such as mathematical reasoning and code-generation tasks. In response:

**1.Challenging Tasks:** Firstly, we demonstrated significant improvements on a harder personalized generation task. Secondly, although mathematical and coding tasks lie significantly beyond our scope and differ from personalized long-text generation, we conducted additional experiments as requested. Our method outperforms the SFT baseline on these tasks. However, fully applying our approach to such domains may require a more advanced optimization pipeline, which represents a promising direction for future work.

**2.Additional point on memory cost:** Consistent with Reviewer 2, this reviewer requested a complete and detailed memory-cost analysis. We have added a comprehensive and formal table clearly demonstrating the efficiency advantages of our method.

---
---
**We have integrated all rebuttal content and additional experiments into the revised manuscript, and highlighted all modifications in blue.**

We hope this summary provides a clear overview of the review and discussion process and assists in your final evaluation.

Warm regards,

Authors of Submission 8572

---

### Meta-Review · Area_Chair_om9Q · 2026-01-09

**Summary:**

This paper proposes a way to do personalized generation from an LLM. The basic way to do this will be to generate a reasoning chain inferring user preference from context and use it to generate. Authors argue that this is inefficient since user preference can change as answer is written. They propose a new way to address this by training a reasoning and generator model where reasoning model generates a single reasoning vector given context and current generation and generator uses the reasoning vector and previous generation and context to generate the response. These two models are called in staggered manner.

Main user concerns involved around seeking more details which authors have done a fairly good job of providing. Authors have added new metrics, new LLM, and provided more hyperparameter and computational analysis.

My main concern with this approach is whether simpler solution would work better, e.g., by prompting an LLM to generate a plan for a given task and then update the plan as the generation proceeds. There is a fair amount of work on plan-based LLM generations (e.g., see Aflow https://arxiv.org/pdf/2410.10762). The proposed approach requires training which makes it more expensive as one cannot use existing LLMs that are being trained with significant monetary investment. Nevertheless, authors have compared with reasonable number of baselines and the idea conceptually makes sense that I am leaning towards weak accept.

**Reviewer Concerns:**

Reviewers raised the following concerns:

1. Reviewer vYSZ asked for more details about relative size of reasoner and generator, sensitivity of proposed approach to context length and interpretability of embeddings. Authors have addressed this.

2. Reviewer uQ75 asked for more evaluation metric, LLMs and mentioned concern over 2x memory requirement. Authors have provided additional experiments for the first two and, in completed form, these should go into the paper. Particularly, all main experiments should have results with an LLM eval metric. Authors mentioned that reasoner can be much cheaper than the generator.

3. Reviewer tgii asked for experiments on coding/maths. Authors have provided a result that is somewhat negative. I think this is very much appreciated since it is important to know the limitations. Authors argued that coding/maths domain are less suited for long-generation personalization. Neverthless, it is surprising that authors see such a big drop in performance with SFT. I believe

**Reviewer Scores:**

1. As Reviewer vYSZ's concerns were addressed, I believe they would have increased their score to 8.

2. As Revewier uQ75's concerns were addressed, they would have increased their score to 6.

3. Reviewer tgii main concern was not addressed and so I believe they would have kept their score at 4.

Overall, borderline paper leaning towards acceptance. Therefore, I recommend weak accept.

---

### Decision · Program_Chairs · 2026-01-26

Accept (Poster)